# Unifying Appearance Codes and Bilateral Grids for Driving Scene Gaussian Splatting

**Nan Wang**[1][†] **Yuantao Chen**[1]**, Lixing Xiao**[1]**, Weiqing Xiao**[1]**, Bohan Li**[3,4]
**Zhaoxi Chen**[1]**, Chongjie Ye**[1]**, Shaocong Xu**[1]**, Saining Zhang**[1]**, Ziyang Yan**[1]
**Pierre Merriaux**[6]**, Lei Lei**[6]**, Tianfan Xue**[5]**, Hao Zhao**[1,2][‡]

[1]BAAI; [2]AIR, THU; [3]SJTU; [4]EIT(Ningbo); [5]CUHK; [6]LeddarTech

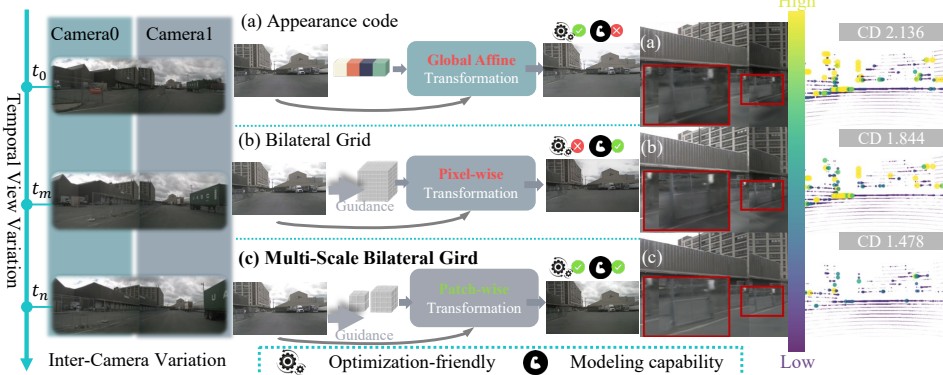

Figure 1: (a) Appearance codes rely on global affine transformations but have limited modeling capability. (b) Bilateral grids perform pixel-wise transformations, improving photometric consistency but are challenging to optimize. (c) The proposed multi-scale bilateral grid unifies appearance codes and bilateral grids, enabling patch-wise transformations.

## Abstract

Neural rendering techniques, including NeRF and Gaussian Splatting (GS), rely on photometric consistency to produce high-quality reconstructions. However, in real-world driving scenarios, it is challenging to guarantee perfect photometric consistency in acquired images. Appearance codes have been widely used to address this issue, but their modeling capability is limited, as a single code is applied to the entire image. Recently, the bilateral grid was introduced to perform pixel-wise color mapping, but it is difficult to optimize and constrain effectively. In this paper, we propose a novel multi-scale bilateral grid that unifies appearance codes and bilateral grids. We demonstrate that this approach significantly improves geometric accuracy in dynamic, decoupled autonomous driving scene reconstruction, outperforming both appearance codes and bilateral grids. This is crucial for autonomous driving, where accurate geometry is important for obstacle avoidance and control. Our method shows strong results across four datasets: Waymo, NuScenes, Argoverse, and PandaSet. We further demonstrate that the improvement in geometry is driven by the multi-scale bilateral grid, which effectively reduces floaters caused by photometric inconsistency. Our code is open-sourced at https://bigcileng.github.io/bilateral-driving.

[†]bigcileng@gmail.com

[‡]Corresponding author: zhaohao@air.tsinghua.edu.cn

39th Conference on Neural Information Processing Systems (NeurIPS 2025).

# 1 Introduction

Neural rendering techniques, such as NeRFs [24, 37, 54, 20, 55] and Gaussian Splatting (GS) [14, 2, 6, 57, 29, 53, 28], have demonstrated significant potential in producing high-quality 3D reconstructions by leveraging photometric consistency. However, ensuring photometric consistency across images in real-world scenarios remains a challenge, as lighting conditions, viewpoints, and camera settings can vary considerably [23, 56, 15, 36, 8]. In autonomous driving applications [46, 41, 35, 5, 17, 13, 34], these challenges are amplified due to the presence of multiple cameras capturing the same scene from different angles over time. As illustrated in the left half of Teaser. 1, we observe that these variations—whether temporal (across different time steps) or inter-camera (across different viewpoints)—can introduce significant discrepancies in image appearance. These inconsistencies pose difficulties in accurately reconstructing dynamic driving environments.

To address this issue, appearance codes [35, 23, 5] have been introduced to encode per-image information and assist in correcting the photometric discrepancies. These codes, while effective, have limitations in modeling capability since they apply a single transformation across the entire image, often neglecting local variations that may occur in complex scenes. For example, in dynamic scenes, such as those encountered in autonomous driving environments, lighting and viewpoints can change across different camera angles and time frames. As shown in the right top of Teaser. 1, a global affine transformation based on a single appearance code may not be sufficient to handle such variations (In Teaser. 1 NuScenes (a), the fence is blurred).

Recent advancements [38, 10] have introduced bilateral grids to perform pixel-wise transformations, enabling improved photometric consistency by allowing more localized adjustments. However, these grids face significant challenges in optimization, as they require complex constraints to avoid instability during training (As shown in Teaser. 1 (b), bilateral grids fail to effectively optimize the reconstruction of large, complex scenes.). In this work, we propose a novel solution—a multi-scale bilateral grid—integrating the strengths of appearance codes and bilateral grids. As shown in Teaser. 1, this new approach allows for patch-wise transformations. Interestingly, in extreme cases, the multi-scale bilateral grid naturally converges to either the bilateral grid or appearance code, depending on the scale. At the finest scale, the multi-scale grid behaves like a traditional bilateral grid, performing pixel-wise transformations to adjust for local variations. In contrast, at the coarsest scale, it effectively reverts to a more global transformation, resembling the behavior of appearance codes. This flexible, scale-dependent approach offers the best of both worlds, providing localized fine-tuning where necessary, while maintaining global consistency when appropriate.

We demonstrate that the multi-scale bilateral grid significantly enhances the geometric accuracy of dynamic, decoupled driving scenes, which is essential for autonomous driving applications where precise geometry plays a crucial role in tasks like obstacle avoidance and path planning. This is evidenced by our evaluation on multiple widely used datasets, including Waymo [30], NuScenes [1], Argoverse [40], and PandaSet [42]. The corresponding Chamfer Distance (CD) values, displayed in the Teaser. 1, highlight a marked reduction in error and improvement in reconstruction quality. Additionally, our method reduces photometric inconsistency, mitigating the appearance of floaters and enhancing the overall realism of the scene.

The contributions of this paper are as follows: **First**, we introduce a novel multi-scale bilateral grid that unifies appearance codes and bilateral grids, transitioning to either of the two paradigms in extreme cases, thus enhancing both modeling capability and optimization efficiency. **Second**, we show that by addressing photometric inconsistencies, it improves the geometric accuracy of dynamic, decoupled driving scene reconstructions. **Third**, we provide extensive benchmarking across four widely used datasets—Waymo, NuScenes, Argoverse, and PandaSet—where our method outperforms previous approaches, showcasing notable improvements in both qualitative and quantitative results.

# 2 Related Works

## 2.1 Bilateral Grids and Appearance Codes

Recent advancements in bilateral grids and appearance codes have significantly influenced techniques for addressing photometric inconsistencies, which are particularly critical in dynamic environments such as autonomous driving. The bilateral grids, first introduced for real-time edge-aware image processing [4], enable efficient manipulation of spatial and intensity variations, forming the foundation

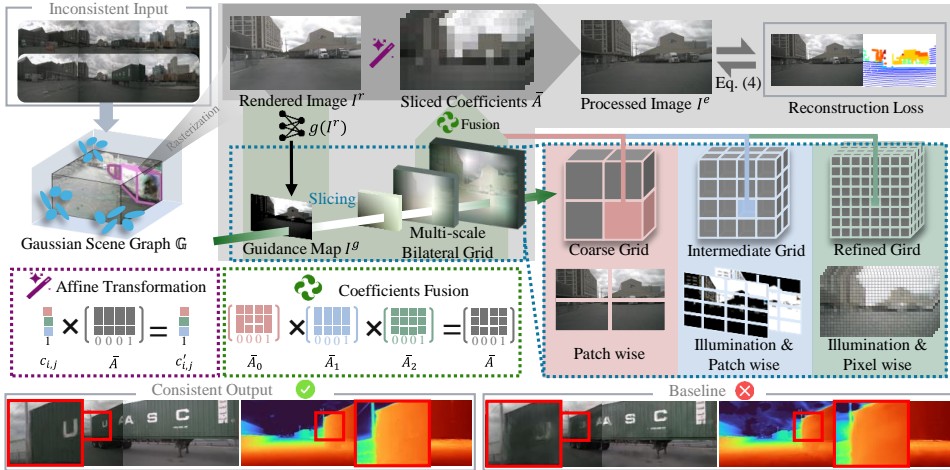

Figure 2: **Overview of our method.** We unify appearance codes with multi-scale bilateral grids. Initially, a coarse rendering is obtained from a Gaussian scene graph. This rendered image is then processed by our multi-scale bilateral grids to perform detailed per-pixel color modeling, guided by a luminance-based map through slice and fusion operations.

for many modern techniques [10, 22, 3, 44, 58]. Extensions like deep bilateral learning [10] further enhance this framework by predicting local affine transformations in bilateral space, achieving real-time image enhancement on resource-constrained devices. Similarly, bilateral-guided upsampling [3] and cost volume refinement [44] have proven effective in tasks ranging from high-dynamic-range imaging to stereo matching, demonstrating the versatility of bilateral representations. In the context of neural rendering, methods such as NeRF [31, 12, 23] and Gaussian Splatting (GS) [56, 15] have achieved groundbreaking results in novel view synthesis but photometric inconsistency caused by varying illumination or transient occluders [23, 56] remains a challenge. Techniques like bilateral guided radiance field processing [38] address these issues by disentangling camera-specific enhancements and reapplying them consistently in 3D space. WildGaussians [15] integrates DINO-based appearance codes into 3D GS to robustly handle occlusions and dynamic lighting in uncontrolled scenes. Cross-Ray NeRF [52] employs cross-ray feature covariance and grid-based transient masking to harmonize appearance variations and suppress occlusions in unconstrained image collections. Despite these strides, existing approaches often struggle to balance global consistency with localized adaptability. For instance, while appearance codes provide global adjustments, they lack the granularity to model fine-grained variations [7]. Conversely, bilateral grids excel at pixel-wise transformations but are challenging to optimize effectively [27]. To address this issue, we introduce a multi-scale bilateral grid which unifies bilateral grids and appearance codes, facilitating high-quality dynamic autonomous driving scene reconstruction with enhanced modeling capability and optimization efficiency.

## 2.2 Autonomous Driving Simulation

Autonomous driving simulation has emerged as a critical tool for developing perception, planning, and control systems by generating diverse, realistic driving scenarios [41, 26, 45, 19, 33, 17, 47, 9]. Recent advancements focus on photorealistic rendering, dynamic scene modeling, and multi-modal sensor simulation [41, 16, 60, 35, 45, 51, 50, 43]. For instance, MARS [41] employs a modular NeRF-based framework to independently control static and dynamic scene elements, while Street Gaussian [45] achieves real-time urban scene rendering using explicit 3D Gaussian representations. DrivingGaussian [60] enhances dynamic scene reconstruction via composite Gaussian Splatting, ensuring occlusion accuracy and multi-camera consistency. NeuRAD [35] integrates sensor-specific effects (*e.g.*, rolling shutter, LiDAR beam divergence) to improve novel view synthesis. Holistic scene understanding and editing have also seen progress. HUGS [59] combines static and dynamic 3D Gaussians for real-time semantic parsing. ChatSim [39] enables language-driven scene editing with external asset integration. LiDAR data integration has advanced through methods like LiDAR-NeRF [32], which uses structural regularization for low-texture regions. HO-Gaussian [18] merges grid-based volumes with Gaussian Splatting to eliminate Structure-from-Motion dependencies.

Despite progress, gaps persist between simulated and real-world data [19, 33, 49, 48]. AlignMiF [33] addresses LiDAR-camera misalignments via geometry-aligned implicit fields, while RodUS [25] decomposes urban scenes into static and dynamic components using 4D semantics to reduce artifacts. Innovations in large-scale generation include InfiniCube [21], which leverages sparse-voxel representations for unbounded dynamic scenes, and Omnire [5], which reconstructs diverse dynamic objects (*e.g.*, pedestrians) for human-vehicle interaction simulations. Furthermore, recent works like GaussianPro [6] and SplatAD [11] explore to refine 3D GS for real-time rendering of dynamic scenes. GaussianPro [6] introduces progressive propagation for texture-less surfaces, and SplatAD [11] models sensor-specific phenomena (*e.g.*, rolling shutter, LiDAR intensity). In this work, we aim to reconstruct high-quality driving scenes by unifying appearance codes and bilateral grids to enhance photometric consistency and modeling capability.

## 3 Methodology

### 3.1 Problem Formulation

We aim to reconstruct a 3D scene representation $\mathbb{G}$ from multi-view images and LiDAR depth maps, commonly available in autonomous driving datasets. Given a set of images $\{I_{c,t}\}$ and corresponding depths $\{D_{c,t}\}$ captured by $N$ cameras over $T$ time steps, we formulate our objective as minimizing the discrepancy between rendered and observed images and depths:

$$\min_{\mathbb{G}} \sum_{c,t} (\left\| I_{c,t}^r - I_{c,t} \right\|_2^2 + \lambda_D \left\| D_{c,t}^r - D_{c,t} \right\|_2^2) \ , \tag{1}$$

where $I_{c,t}^r$ and $D_{c,t}^r$ are the rendered appearance and geometry of the scene $\mathbb{G}$ from camera $c$ at time $t$, and $\lambda_D$ is a weight balancing the depth term.

However, dynamic driving scenes and varying camera properties cause inconsistent appearances, leading to geometric and texture artifacts. To address inconsistency, we decompose each image $I_{c,t}$ into consistent $\mathcal{C}_{c,t}$ (*e.g.*, intrinsic scene colors and constant sensor adjustments like normalization exposure), and non-consistent $\mathcal{N}_{c,t}$ (*e.g.*, components-varying lighting conditions, different camera settings and diverse image ISP effects):

$$I_{c,t} = \mathcal{C}_{c,t} + \mathcal{N}_{c,t} \ , \tag{2}$$

We model $\mathcal{N}_{c,t}$ as a non-linear transformation $\mathcal{F}(\mathcal{C}_{c,t})$. Substituting into Eq. (2):

$$I_{c,t} = \mathcal{C}_{c,t} + \mathcal{F}(\mathcal{C}_{c,t}) \ , \tag{3}$$

Let $\mathcal{E}(\mathcal{C}_{c,t}) = \mathcal{C}_{c,t} + \mathcal{F}(\mathcal{C}_{c,t})$, we can reformulate the optimization objective in Eq. (1) as:

$$\min_{\mathbb{G}} \sum_{c,t} (\left\| \mathcal{E}(\mathcal{C}_{c,t}^r) - I_{c,t} \right\|_2^2 + \lambda_D \left\| D_{c,t}^r - D_{c,t} \right\|_2^2) \ , \tag{4}$$

This reformulation links photometric and geometric consistency optimization within a joint objective, guiding $\mathcal{C}_{c,t}^r$ adjustments by geometric constraints ($D_{c,t}^r$). This mitigates texture-geometry ambiguities (e.g., shadows on road surfaces misinterpretable as geometry changes). $\mathcal{F}(\mathcal{C}_{c,t})$ models transient appearance, while $\mathcal{C}_{c,t}^r$ enforces consistent scene properties. Experiments (Sec. 4, Tab. 1) show this joint optimization reduces both photometric error and geometric drift in comparison to baselines.

### 3.2 Gaussian Splatting for Autonomous Driving Environments

Following prior methodologies [5, 35], we represent autonomous driving environments using a hybrid scene graph, decomposing the scene into sky, background, and dynamic object models. Each dynamic object is represented by a 3D model in canonical space and transformed to the scene with an associated sequence of SE(3) transformations. We derive the transformation matrixes from existing object detection pipelines or ground-truth annotations.

For the sky, we use an environment map to model sky color based on viewing direction, while static background is represented as a set of semi-transparent 3D Gaussians. Each Gaussian is characterized by a learnable opacity parameter $o \in (0, 1)$, a mean position $\mu \in \mathbb{R}^3$, and an anisotropic covariance matrix $\Sigma \in \mathbb{R}^{3 \times 3}$ which is parameterized by a scale vector $S \in \mathbb{R}^3$ and a rotation quaternion $q \in \mathbb{R}^4$. Additionally, Spherical harmonics coefficients $c \in \mathbb{R}^F$ are used to model appearance.

For the moving object models, we further distinguish between non-deformable (*e.g.*, vehicles) and deformable objects (*e.g.*, pedestrians). Non-deformable objects $\{\mathbb{G}_i^N | i \in \{1, \ldots, n_n\}\}$ are optimized in their local coordinate space and transformed into the global world space via their pose $T$:

$$\mathbb{G}_i^N = T \otimes \hat{\mathbb{G}}_i^N \ , \tag{5}$$

For deformable objects $\{\mathbb{G}_i^D | i \in \{1, \ldots, n_d\}\}$, we employ a deformation network $\mathcal{F}_\Phi$ (parameterized by $\Phi$) to adapt the Gaussian representation based on latent variable $e$ and time $t$:

$$\mathbb{G}_i^D = T \otimes \left( \hat{\mathbb{G}}_i^D \oplus \mathcal{F}_\Phi(\hat{\mathbb{G}}_i^D, e, t) \right) \ . \tag{6}$$

### 3.3  Multi-Scale Bilateral Grid for Appearance Enhancement

Solving the photometric correction problem formulated in Eq. (4) fundamentally depends on the ability to effectively model the complex and spatially-varying photometric transformations present in real-world driving scenes. The function $\mathcal{E}(\cdot)$ in our formulation represents this crucial photometric enhancement process. To address this, we propose the Multi-Scale Bilateral Grid architecture, detailed in this section, as a novel grounded solution for approximating $\mathcal{E}(\cdot)$ and achieving high-quality, consistent appearance enhancement.

#### 3.3.1  Multi-Scale Bilateral Grid Architecture

To address limitations of existing methods in handling diverse photometric variations in driving scenes, we propose a multi-scale bilateral grid framework. Our framework achieves this unification by employing a hierarchical pyramid of bilateral grids, organized across multiple scales. This multi-scale design is directly inspired by the nature of photometric inconsistencies in real-world environments, which range from global scene-level changes to highly localized variations (*e.g.*, from overall lighting shifts to fine texture-level shadows). By utilizing this grid hierarchy, our framework aims to capture and effectively correct photometric variations at their corresponding scales, thus enabling a more comprehensive and spatially adaptable photometric correction.

The multi-scale bilateral grid transformation can be formulated as $I^e = \bar{A} \odot I^r$, where $\bar{A}$ represents a composite, scale-dependent photometric transformation. Crucially, this composite transformation is constructed hierarchically, combining transformations learned by individual bilateral grids at different scales, progressing from coarse to fine. This staged composition is key to achieving scale-dependent adaptation to photometric inconsistencies and allows us to move beyond the inherent limitations of single-scale methods. Further details are provided in the subsequent sections.

Our framework achieves this hierarchical representation using a three-level bilateral grid pyramid (Fig. 2): **(1) Coarse Level** ($2 \times 2 \times 1 \times 12$ grid) captures global scene structure and approximate global appearance codes; **(2) Intermediate Level** ($4 \times 4 \times 2 \times 12$ grid) represents regional features and capture mid-range photometric variations; **(3) Fine Level** ($8 \times 8 \times 4 \times 12$ grid) encodes local details and approximate pixel-wise bilateral grids within the multi-scale framework.

At each level $l$, the grid tensor is defined as $\mathcal{A}^{(l)} \in \mathbb{R}^{H^{(l)} \times W^{(l)} \times D^{(l)} \times C^{(l)}}$, where Spatial Dimension $(H^{(l)}, W^{(l)})$ represents spatial resolution (height, width) at level $l$, Guidance Dimension $D^{(l)}$ specifies the guidance intensity levels at level $l$ and Coefficient Channel $C = 12$ represents a flattened $3 \times 4$ affine color transformation matrix.

#### 3.3.2  Guidance Map, Slice Operation, and Multi-Scale Fusion

To achieve adaptive and spatially-varying photometric correction, our multi-scale bilateral grid framework employs a guidance map and a *slicing* operation [4] to retrieve localized transformations, followed by a hierarchical fusion strategy to combine transformations across scales.

We first derive a luminance-based guidance map $I^g(u, v) = \text{GrayScale}(I^r(u, v))$ from the rendered image $I^r(u, v)$, following the approach of [4, 38, 10]. This map encodes spatial brightness variations like shadows and highlights, serving as a spatially-varying query to our multi-scale grid. For each level $l$ and pixel $(u, v)$, the luminance $d$ from $I^g$ is used to perform a *slicing* operation, querying the grid's intensity dimension $D^{(l)}$ to locate an intensity bin. Affine transformation coefficients $\mathcal{A}_{i,j,k}^{(l)}$ around this bin are then combined via trilinear interpolation for a level-specific transformation

$\bar{A}^{(l)}(u, v)$. The *slicing* operation retrieves per-pixel transformations from each grid level, written as:

$$\bar{A}^{(l)}(u, v) = \sum_{i,j,k} w_{i,j,k}(u, v, d) \mathcal{A}^{(l)}(i, j, k) \ , \tag{7}$$

To efficiently fuse transformations across scales, we utilize hierarchical function composition. A naive approach of full-resolution *slicing* at each scale is computationally expensive and redundant due to local photometric coherence. To address this, we employ downsampled guidance maps $I^{g(l)}$ for each scale $l$, performing *slicing* to obtain low-resolution coefficient fields, which are then upsampled. This approach explicitly links scale-aware guidance to patch operations-the reduced-resolution maps enable efficient spatial aggregation of photometric transformations while enhances efficiency (See ablation study in Tab. 5). The level-specific transformations $\mathcal{T}^{(l)}$, decomposed into linear matrices $\bar{M}^{(l)}$ and translations $\bar{T}^{(l)}$, are then sequentially composed from coarse to fine to produce the final composite transformation $\bar{A}$. The refined image $I^e(u, v)$ is obtained by applying $\bar{A}(u, v)$ to $I^r(u, v)$, with the fusion process expressed as:

$$I^e = \mathcal{T}^{(L-1)} \circ \mathcal{T}^{(L-2)} \circ \cdots \circ \mathcal{T}^{(0)}(I^r) \ , \tag{8}$$

This hierarchical fusion decomposes the photometric transformation into scale-dependent residual refinements. Coarse scales capture global transformations, intermediate scales refine regional variations, and fine scales address local details (Further elaboration in Sec. 4.3). Further details on the interpolation kernel and computational efficiency considerations are provided in Appendix A1.1.

Table 1: Ablation studies and comparisons on four large-scale driving datasets. Our proposed method consistently outperforms OmniRe across all metrics and datasets. **w/AC** denotes appearance codes; **w/BG** denotes single bilateral grids (size of $16 \times 16 \times 8$).

| Dataset | Method | Reconstruction | | | Novel View Synthesis | | | Geometry | | |
|---|---|---|---|---|---|---|---|---|---|---|
| | | PSNR ↑ | SSIM ↑ | LPIPS ↓ | PSNR ↑ | SSIM ↑ | LPIPS ↓ | CD ↓ | RMSE ↓ | Depth ↓ |
| Waymo | OmniRe | 28.92 | 0.833 | 0.295 | 26.40 | 0.761 | 0.311 | 1.482 | 2.785 | 0.540 |
| | OmniRe w/AC | 28.95 | 0.835 | 0.293 | 26.44 | 0.759 | 0.315 | 1.378 | 2.760 | 0.519 |
| | OmniRe w/BG | 28.19 | 0.831 | 0.292 | 21.51 | 0.743 | 0.333 | 1.523 | 2.798 | 0.492 |
| | Ours | **29.23** | **0.836** | **0.289** | **26.55** | **0.762** | **0.310** | **0.989** | **2.744** | **0.477** |
| NuScenes | OmniRe | 26.37 | 0.837 | 0.209 | 23.74 | 0.733 | 0.232 | 1.458 | 3.420 | 0.110 |
| | OmniRe w/AC | 26.38 | 0.840 | 0.204 | 23.74 | 0.732 | 0.229 | 1.437 | 3.415 | 0.106 |
| | OmniRe w/BG | 25.98 | 0.837 | 0.209 | 23.60 | 0.705 | 0.262 | 1.380 | 3.390 | 0.097 |
| | Ours | **27.69** | **0.847** | **0.193** | **24.64** | **0.739** | **0.216** | **1.161** | **3.340** | **0.059** |
| Argoverse | OmniRe | 24.59 | 0.848 | 0.202 | 22.53 | 0.755 | 0.220 | 0.954 | 4.208 | 0.050 |
| | OmniRe w/AC | 24.58 | 0.848 | 0.201 | 22.51 | 0.756 | 0.219 | 0.959 | 4.215 | 0.051 |
| | OmniRe w/BG | 23.31 | 0.842 | 0.216 | 21.70 | 0.725 | 0.254 | 0.901 | 4.218 | 0.049 |
| | Ours | **24.68** | **0.849** | **0.200** | **22.58** | **0.756** | **0.217** | **0.807** | **4.199** | **0.040** |
| Pandaset | OmniRe | 30.20 | 0.903 | 0.219 | 27.49 | 0.835 | 0.240 | 0.503 | 2.874 | 0.018 |
| | OmniRe w/AC | 30.20 | 0.903 | 0.220 | 27.51 | 0.841 | 0.242 | 0.496 | 2.871 | 0.020 |
| | OmniRe w/BG | 29.73 | 0.904 | 0.220 | 27.38 | 0.830 | 0.246 | 0.484 | 2.867 | 0.013 |
| | Ours | **30.75** | **0.906** | **0.213** | **27.89** | **0.847** | **0.235** | **0.453** | **2.852** | **0.011** |

## 4 Experiments

### 4.1 Experimental Setup

**Datasets**. We evaluate on four autonomous driving datasets: Waymo [30], NuScenes [1], Argoverse [40], and PandaSet [42], selected for their diversity in sensor configurations (LiDAR, camera specifications), environmental conditions (lighting, seasons), and geographic locations. Identical model hyperparameters are used across all datasets to test generalization capability. Technical specifications including camera counts and ego-vehicle view cropping details are provided in Appendix A2.

**Baseline**. Our focus is on modeling appearance variance across viewpoints, we build upon the open-source works *DriveStudio* [5], *ChatSim* [39], *StreetGS* [45] and test three approaches: appearance codes, standalone bilateral grids, and our multi-scale bilateral grids. This allows for direct comparison of how each method handles photometric inconsistencies across viewpoints.

**Implementation details**. We evaluate our method on many autonomous driving datasets to demonstrate its robustness. For each dataset, We train all the methods on a single NVIDIA L40 GPU and

Table 2: **Comparison with different baseline approaches for scene reconstruction.** Performance is reported as the average over the nuScenes [1] scenarios.

| Method | Reconstruction | | | Novel View Synthesis | | | Geometry | | |
|---|---|---|---|---|---|---|---|---|---|
| | PSNR ↑ | SSIM ↑ | LPIPS ↓ | PSNR ↑ | SSIM ↑ | LPIPS ↓ | CD ↓ | RMSE ↓ | Depth ↓ |
| ChatSim [39] | 25.10 | 0.805 | 0.252 | 23.27 | 0.725 | 0.270 | 1.557 | 3.509 | 0.106 |
| ChatSim (Ours) | **27.04** | **0.819** | **0.231** | **24.67** | **0.735** | **0.249** | **1.236** | **3.412** | **0.053** |
| StreetGS [45] | 25.74 | 0.822 | 0.240 | 23.64 | 0.736 | 0.259 | 1.604 | 3.544 | 0.107 |
| StreetGS (Ours) | **27.90** | **0.836** | **0.219** | **25.10** | **0.747** | **0.238** | **1.272** | **3.458** | **0.055** |

compare the 3D reconstruction and novel view synthesis (NVS) results between camera and LiDAR data. Additionally, we ablate key components of our method and quantify their impact on both appearance and geometry quality.

**Training and Dynamic Rendering**.

**1) Training.** To optimize our multi-scale Gaussian scene representation, we employ a joint training strategy that minimizes a composite reconstruction loss :

$$\mathcal{L}_{recon} = \lambda_r \mathcal{L}_1 + \lambda_s \mathcal{L}_{\text{SSIM}} + \lambda_d \mathcal{L}_d + \lambda_o \mathcal{L}_o \ , \tag{9}$$

Furthermore, we introduce two regularization terms to enhance image fidelity: *Adaptive Total Variation Regularization*. This term encourages smoothness and reduces noise while preserving image details, and *Circle Regularization Loss*. This loss applies inverse transformation to the ground-truth images, preventing discrepancies and image quality degradation. As visually demonstrated in Fig. 5, these terms effectively improve image fidelity. Detailed are provided in Appendix A1.

**2) Rendering.** For dynamic rendering using our multi-scale bilateral grid framework, we employ a specific interpolation strategy when encountering novel test images, especially those simulating dynamic ISP conditions. We first conducts a temporal proximity search to identify the most relevant grids. Following this, we perform scale-specific interpolation using the two nearest grids found.

Table 3: **Evaluation on Challenging Scene Subsets.** Results on systematically selected extreme cases. Our method shows substantially improvements when photometric inconsistencies are severe.

| Scene Type | Method | Reconstruction | | | Novel View Synthesis | | | Geometry | | |
|---|---|---|---|---|---|---|---|---|---|---|
| | | PSNR ↑ | SSIM ↑ | LPIPS ↓ | PSNR ↑ | SSIM ↑ | LPIPS ↓ | CD ↓ | RMSE ↓ | Depth ↓ |
| Challenging Scenes | OmniRe | 26.84 | 0.847 | 0.239 | 24.46 | 0.758 | 0.263 | 0.650 | 3.176 | 0.063 |
| (18 scenes) | **Ours** | **28.13** | **0.856** | **0.223** | **25.17** | **0.763** | **0.250** | **0.523** | **3.120** | **0.023** |
| Night Scenes | OmniRe | 28.34 | 0.805 | 0.338 | 26.54 | 0.753 | 0.360 | 0.586 | 2.767 | 0.041 |
| (4 scenes) | **Ours** | **29.13** | **0.812** | **0.330** | **27.21** | **0.762** | **0.349** | **0.500** | **2.731** | **0.016** |
| Sun Flare Scenes | OmniRe | 26.68 | 0.866 | 0.195 | 24.98 | 0.796 | 0.212 | 0.781 | 3.073 | 0.079 |
| (5 scenes) | **Ours** | **28.65** | **0.879** | **0.173** | **25.84** | **0.803** | **0.192** | **0.593** | **2.994** | **0.022** |

## 4.2 Quantitative Evaluation

**Reconstruction and Geometric Quality.** We present a comprehensive quantitative analysis in Tab. 1, comparing our full model against the OmniRe and its variants with appearance codes (w/AC) and a single bilateral grid (w/BG). Our method consistently sets a new state-of-the-art across all four datasets in both appearance and, most critically, geometric metrics.

In terms of appearance fidelity, our model achieves the highest PSNR and SSIM scores. For example, on NuScenes, our method reaches a PSNR of 27.69, a significant jump from 26.37 (OmniRe) and 26.38 (OmniRe w/AC). This demonstrates the superior ability of the multi-scale grid to model complex photometric variations.

However, the most striking result is the substantial improvement in geometric accuracy. Our method drastically reduces the Chamfer Distance, a key indicator of geometric fidelity. On the Waymo dataset, our model hits a CD of 0.989, outperforming the best baseline (OmniRe w/AC, 1.378) by a remarkable 28.2%. This trend holds across all datasets, with significant reductions in **Chamfer Distance (CD)**, **Root Mean Squared Error (RMSE)**, and **Median Squared Depth Error (Depth)**. This result provides strong evidence for our central hypothesis: by accurately resolving photometric ambiguities, our multi-scale approach effectively eliminates geometric artifacts like "floaters" and

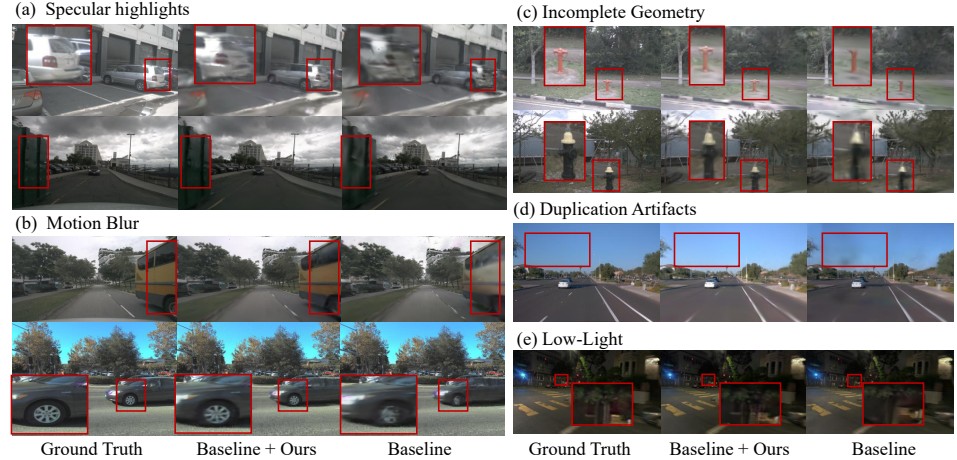

| (a) Specular highlights | (c) Incomplete Geometry |
| (b) Motion Blur | (d) Duplication Artifacts |
| | (e) Low-Light |

| Ground Truth | Baseline + Ours | Baseline | Ground Truth | Baseline + Ours | Baseline |

Figure 3: Our Method vs. Baseline Methods on Waymo, NuScenes, Argoverse, and PandaSet.

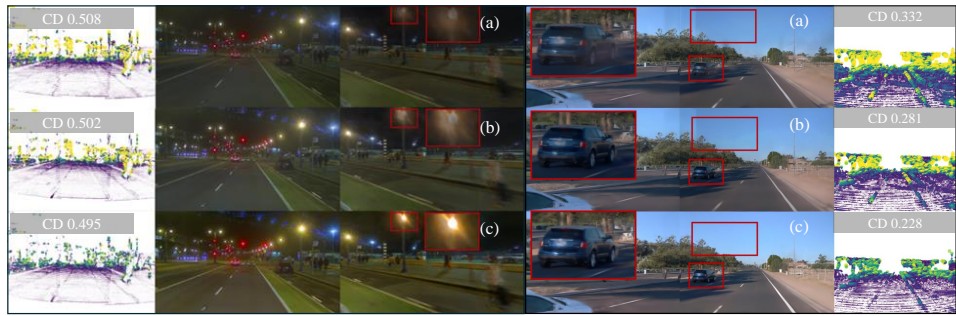

Figure 4: Our proposed framework (c) outperforms appearance codes (a) and single bilateral grids (b) by simultaneously addressing optimization challenges and enhancing geometric modeling. This delivers significant geometric accuracy improvements (lower Chamfer Distance (CD) via multi-scale bilateral grid, reducing error and improving reconstruction) and reduced photometric inconsistency (fewer floaters) in dynamic, decoupled driving scenes. Yellow indicates a high lidar error, while Purple indicates a low lidar error.

produces a much cleaner, more accurate 3D reconstruction. The single bilateral grid (w/BG), despite its high expressiveness, often fails to converge and even degrades performance, highlighting the optimization challenges that our multi-scale, coarse-to-fine framework successfully overcomes.

**NVS.** As shown in Tab. 1, our model's superior reconstruction quality translates directly to improved NVS. Our method consistently outperforms all baselines in NVS metrics across the four datasets. For instance, on PandaSet, we achieve a PSNR of 27.89, surpassing the next best (OmniRe w/AC at 27.51). This indicates that the cleaner geometry and more consistent appearance learned by our model generalize better to unseen viewpoints, producing higher-fidelity renderings.

**Generalization to Other Methods.** To demonstrate the broad applicability of our approach, we integrated our multi-scale bilateral grid into two other state-of-the-art methods, ChatSim [39] and StreetGS [45]. The results on NuScenes, presented in Tab. 2, are compelling. Our module brings substantial gains to both frameworks. For StreetGS, incorporating our method boosts the reconstruction PSNR from 25.74 to 27.90 and slashes the Chamfer Distance from 1.604 to 1.272. These significant improvements underscore that our multi-scale grid is not just a bespoke enhancement for one architecture but a generalizable and powerful module for improving photometric and geometric consistency in Gaussian Splatting-based scene reconstruction.

**Performance on Challenging Scenarios**. To validate robustness, we curated a subset of 18 challenging scenarios with extreme photometric inconsistencies. As shown in Table 3, our method's advantages are substantially more pronounced here than on the standard datasets. Across this entire

subset, our model achieves a +1.29 dB PSNR gain. This improvement is even more significant in specific difficult sub-categories; for instance, in "Sun Flare Scenes," the gain reaches +1.97 dB. This demonstrates our model's strong effectiveness in resolving severe, real-world edge cases.

## 4.3 Qualitative Evaluation

**Rendering Results.** Fig. 3 and Fig. 4 present visualization results of our method on scenes from Waymo [30], NuScenes [1], Argoverse [40], and PandaSet [42], demonstrating its capability to handle diverse conditions that might otherwise lead to inconsistencies.

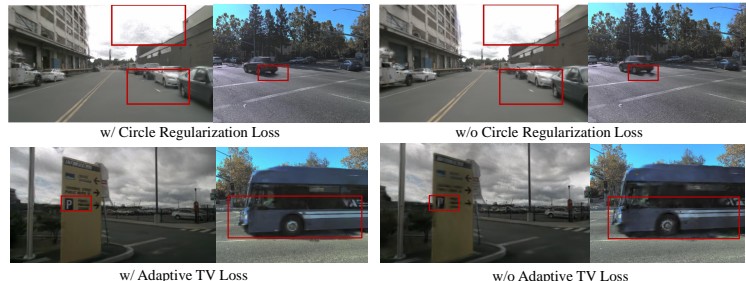

Figure 5: Visualizations of ablations on Circle Regularization loss and Adaptive TV loss.

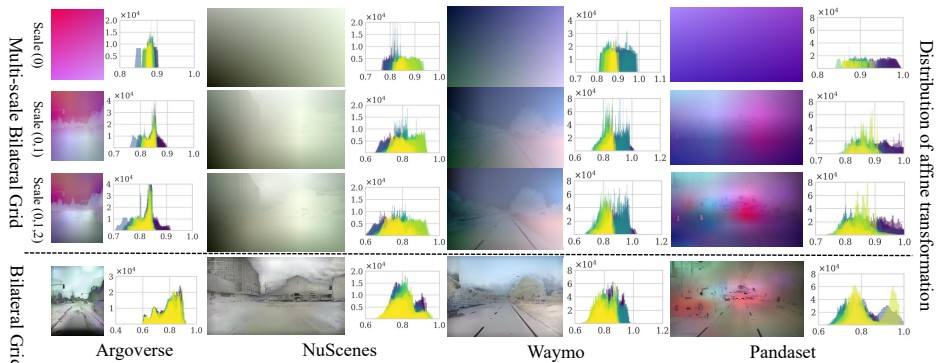

Figure 6: **Visualization of Affine Transformations and Distributions.** The first three rows display the affine transformation and corresponding distributions when applying the first level, the first two levels, and all three levels of our multi-scale approach. The final row compares these to results from a single bilateral grid, demonstrating that the multi-scale bilateral grid achieves a coarse-to-fine, smooth image enhancement with more diverse processing capabilities.

**Analysis of Learned Affine Transformations.** To gain deeper insight into how our model functions, we analyze the distribution of the learned affine transformations in Fig. 6. The histograms for the original single-scale bilateral grid (BG) exhibit a distinct bi-modal, peaky

| Configuration | Grid Params (M) | Training Time (h) | Testing time (FPS) |
|---|---|---|---|
| OmniRe w/AC | 0.072 | 1.93 | 53 |
| OmniRe w/BG | 27.843 | 2.85 | 54 / 38† |
| Ours | 3.969 | 2.10 | 54 / 42† |

Table 4: **Computational Efficiency Analysis.** † denotes rendering with bilateral grid processing active.

distribution for each viewpoint. This suggests that the single grid learns a limited set of dominant photometric corrections for each view, struggling to capture the full spectrum of variations. In stark contrast, the aggregated histogram from our multi-scale grid (across all levels and viewpoints) is significantly flatter and more dispersed. This is a direct visualization of our framework's core strength. The coarse-level grid first establishes a view-dependent baseline appearance, capturing the dominant global photometric shift (interestingly, its own histogram is also peaky, but varies across views). Then, the medium and fine grids learn residual transformations in a coarse-to-fine manner,

correcting the errors from the preceding levels. This hierarchical, residual refinement process allows the model to represent a much broader and more diverse range of photometric transformations. The flatter histogram is empirical evidence of this enhanced representational power, which is crucial for achieving consistent, high-quality renderings across the diverse and challenging views encountered in autonomous driving.

## 4.4 Ablation Studies

As shown in Tab. 4, Tab. 5 and Fig. 5, we systematically evaluate component contributions through:

**Guidance Map Resolution.** We test different downsampling factors for the guidance map at each grid level. The results in Tab. 5(a) show that a moderate downsampling of (2,2,1) from coarse to fine yields the best balance, achieving the highest NVS PSNR (24.64). Using the full resolution (1,1,1) slightly degrades performance, likely due to overfitting to noise, validating our patch-based aggregation strategy.

**Grid Size Combinations.** The combination of grid sizes is crucial. As shown in Tab. 5(b), removing the coarsest grid level (e.g., using only (4,4,2)+(8,8,4)) leads to a dramatic increase in CD, highlighting the importance of the coarse grid for establishing global geometric consistency. Crucially, a baseline with only a single fine-resolution grid (8,8,4) performs poorly, confirming that finer grids fail to converge without the stable, coarse-level initialization our method provides.

| Settings | Recon.↑ | NVS.↑ | CD↓ |
|---|---|---|---|
| Full model | **27.69** | 24.64 | **1.161** |
| *(a) Guidance Map Resolution* | | | |
| (1,1,1) | 27.62 | 24.34 | 1.271 |
| (2,2,1) | 27.68 | **24.64** | 1.170 |
| (8,8,4) | 27.52 | 24.53 | 1.187 |
| (16,16,8) | 27.50 | 24.54 | 1.202 |
| *(b) Grid Size Combinations* | | | |
| Single Grid (8,8,4) | 26.56 | 23.90 | 1.376 |
| (4,4,2)+(8,8,4) | 27.49 | 24.56 | 1.963 |
| (2,2,1)+(8,8,4) | 27.57 | 24.57 | 1.940 |
| (2,2,1)+(4,4,2) | 27.33 | 24.42 | 1.213 |
| *(c) Loss Functions* | | | |
| w/o circle regularization loss | 27.47 | 24.52 | 1.164 |
| w/o adaptive total variance loss | 27.64 | 24.61 | 1.169 |

Table 5: **Ablations.** Numbers in *Guidance Map Resolution* indicate downsample factors from coarse to fine level, and numbers in *Grid Size Combinations* indicate grid size.

**Loss Functions.** Ablating our proposed regularization losses in Tab. 5(c) shows their contribution. Removing the circle regularization loss slightly degrades both reconstruction and NVS performance. The effect of the adaptive total variance loss is more subtle in metrics but visibly improves rendering quality by reducing noise, as shown in Fig. 5.

**Efficiency.** As detailed in Tab. 4, our method strikes an effective balance between performance and computational cost. The single large bilateral grid (w/BG) leads to a massive parameter count (27.8M) and the longest training time (2.85h). In contrast, our multi-scale approach requires only 3.9M parameters and a modest 9% increase in training time (2.10h vs. 1.93h) compared to the appearance code baseline, while delivering substantially better results. During inference, our method maintains high frame rates, demonstrating its practicality.

## 5 Conclusion

We introduced a novel multi-scale bilateral grid that unifies global appearance codes and pixel-wise bilateral grids into a single, hierarchical framework. Our extensive experiments demonstrate that this approach not only effectively models and corrects complex photometric inconsistencies in autonomous driving scenes but, more importantly, leads to a significant and crucial improvement in the geometric accuracy of the final reconstruction. By establishing this strong link between advanced appearance modeling and geometric fidelity, our work paves the way for more robust and reliable neural rendering systems in safety-critical, real-world applications.

**Limitations.** While our method shows significant improvements, some limitations remain. (1) The computational overhead, though much lower than a single large bilateral grid, is still higher than simple appearance codes, which could be a consideration for resource-constrained scenarios. (2) Modeling extremely fast-moving or highly non-rigid objects where LiDAR and camera data misalign remains an open challenge.

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

# Appendix: Unifying Appearance Codes and Bilateral Grids for Driving Scene Gaussian Splatting

## A   Additional Implementation Details

### A.1   Loss Functions and Optimization

We jointly optimize our multi-scale Gaussian scene representation by minimizing a composite loss function $\mathcal{L}_{\text{total}}$:

$$\mathcal{L}_{\text{total}} = \mathcal{L}_{\text{recon}} + \lambda_{\text{TV}}\mathcal{L}_{\text{TV}} + \lambda_{\text{circle}}\mathcal{L}_{\text{circle}} \tag{A1}$$

where $\mathcal{L}_{\text{recon}}$ is the primary reconstruction loss, and $\mathcal{L}_{\text{TV}}$ and $\mathcal{L}_{\text{circle}}$ are regularization terms.

#### A.1.1   Reconstruction Loss

The core reconstruction loss, $\mathcal{L}_{\text{recon}}$, drives the accurate reproduction of both RGB and depth information:

$$\mathcal{L}_{\text{recon}} = \lambda_r\mathcal{L}_1 + (1 - \lambda_r)\mathcal{L}_{\text{SSIM}} + \lambda_d\mathcal{L}_d + \lambda_o\mathcal{L}_o \tag{A2}$$

Here, $\mathcal{L}_1$ and $\mathcal{L}_{\text{SSIM}}$ measure the image-space difference. $\mathcal{L}_d$ represents the loss between the rendered depth and the ground truth LiDAR depth. $\mathcal{L}_o$ is an opacity regularization term that encourages alignment with a non-sky mask.

#### A.1.2   Adaptive Total Variation (TV) Regularization

To encourage smoothness and reduce noise in the optimized grid representations, we incorporate a TV regularization term, $\mathcal{L}_{\text{TV}}$, at each level $l$ of the multi-scale representation:

$$\mathcal{L}_{\text{TV}} = \sum_l k^{(l)} \cdot \frac{1}{|\mathcal{A}^{(l)}|} \sum_{i,j,k} \sum_{\mathbb{D}\in\{x,y,z\}} \left\| \Delta_{\mathbb{D}}\mathcal{A}^{(l)}(i,j,k) \right\|_2^2 \tag{A3}$$

The adaptive weight $k^{(l)}$ for each level $l$ is proportional to the grid size, applying stronger smoothing to finer, higher-resolution grids and lighter regularization to coarser grids.

$$k^{(l)} = a\sqrt{H^{(l)} \cdot W^{(l)} \cdot D^{(l)}} + b \tag{A4}$$

#### A.1.3   Circle Regularization for Photometric Consistency

To constrain the noise introduced by photometric corrections and prevent overly aggressive alterations, we introduce a circle regularization loss, $\mathcal{L}_{\text{circle}}$:

$$\mathcal{L}_{\text{circle}} = \sum_{(u,v)\in\mathcal{S}} \left\| I^r(u,v) - \bar{A}^{-1}(I^{gt}(u,v)) \right\|_2^2 \tag{A5}$$

where $\mathcal{S}$ denotes the set of pixel coordinates. This loss encourages the rendered image $I^r$ to be reconstructible from the ground truth image $I^{gt}$ via an inverse appearance transformation $\bar{A}^{-1}$.

#### A.1.4   Coarse-to-Fine Optimization Strategy

We employ a coarse-to-fine optimization strategy by utilizing level-dependent learning rates (e.g., $1 \times 10^{-5}, 3 \times 10^{-5}, 1 \times 10^{-4}$ from coarse to fine). Coarser grids, assigned higher learning rates, rapidly learn the global scene illumination, while finer grids, with lower learning rates, hierarchically refine high-frequency photometric details. This staged optimization enhances stability.

### A.2   Dynamic Rendering for Real-World ISP Adaptation

To bridge the domain gap between the fixed ISP used during training and dynamic real-world ISP pipelines encountered during inference, we employ an interpolation strategy. For a novel test image with timestamp $t_{\text{novel}}$, we proceed as follows:

1. **Temporal Proximity Search:** We identify the two training timestamps from the same camera, $t_1$ and $t_2$, that are temporally closest to $t_{\text{novel}}$ (such that $t_1 \leq t_{\text{novel}} \leq t_2$).

2. **Scale-Specific Grid Interpolation:** We perform linear interpolation specifically on the **coarse** and **medium-scale** bilateral grids ($\mathcal{A}^{(0)}, \mathcal{A}^{(1)}$) derived from these two timestamps.

The interpolated grid $\hat{\mathcal{A}}^{(l)}$ for $l \in \{0, 1\}$ is formulated as:

$$\hat{\mathcal{A}}^{(l)} = \omega \mathcal{A}^{(l)}_{t_1} + (1 - \omega)\mathcal{A}^{(l)}_{t_2} \tag{A6}$$

The temporal interpolation weight, $\omega$, is determined by proximity:

$$\omega = \frac{t_2 - t_{novel}}{t_2 - t_1} \tag{A7}$$

The fine-scale grid ($l = 2$) is not used during interpolation, as it is primarily intended to capture scene-intrinsic details rather than global ISP variations.

# B  Dataset Details

This section provides dataset-specific details regarding our evaluation protocol, including sequence IDs for reproducibility.

## B.1  Waymo Open Dataset [2]

We use all five cameras and all LiDAR sensors. We conduct our experiments on the following 6 sequences, selected according to [3, 4]:

- segment-10017090168044687777
- segment-10061305430875486848
- segment-10584247114982259878
- segment-15090871771939393635
- segment-4458730539804900192
- segment-5835049423600303130

## B.2  NuScenes [1]

We utilize all six available cameras and all LiDAR sensors. We select the following 8 sequences: 152, 164, 171, 200, 209, 359, 529, 916, which is an extension of the dataset used by [3]. To address ego-vehicle visibility, we crop the bottom 80 pixels from the back camera images.

## B.3  PandaSet [6]

We utilize six cameras and one LiDAR unit. We specifically evaluate on the following 10 challenging nighttime sequences: 063, 066, 070, 073, 074, 077, 078, 079, 088, 149. To mitigate ego-vehicle artifacts, we apply a bottom crop of 260 pixels to the back camera images.

## B.4  Argoverse2 [5]

We leverage the seven ring cameras and both LiDAR sensors available in the dataset. We conduct our experiments on the following 9 sequences, in line with [3]:

- 05fa5048-f355-3274-b565-c0ddc547b315
- 0b86f508-5df9-4a46-bc59-5b9536dbde9f
- 185d3943-dd15-397a-8b2e-69cd86628fb7
- 25e5c600-36fe-3245-9cc0-40ef91620c22
- 27be7d34-ecb4-377b-8477-ccfd7cf4d0bc
- 280269f9-6111-311d-b351-ce9f63f88c81
- 2f2321d2-7912-3567-a789-25e46a145bda
- 44adf4c4-6064-362f-94d3-323ed42cfda9
- 5589de60-1727-3e3f-9423-33437fc5da4b

To minimize ego-vehicle interference, we apply a bottom crop of 250 pixels to the front center, rear left, and rear right camera views.

# C   Additional Quantitative Results

This section presents a comprehensive breakdown of our model's performance, including per-scene metrics, integration results with SOTA methods, and robustness checks on challenging subsets.

## C.1   Per-Scene Geometry and Appearance Evaluation

We provide detailed metrics for geometry (Tab. A1 - A4) and appearance (Tab. A5 - A8).

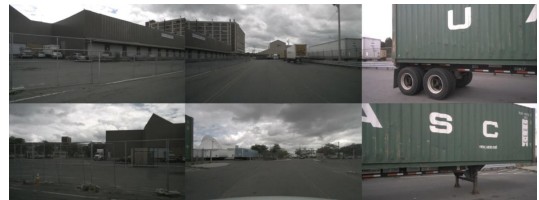

Ground Truth

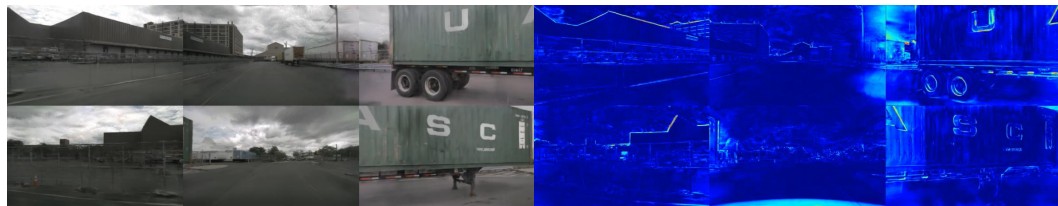

Baseline+AP

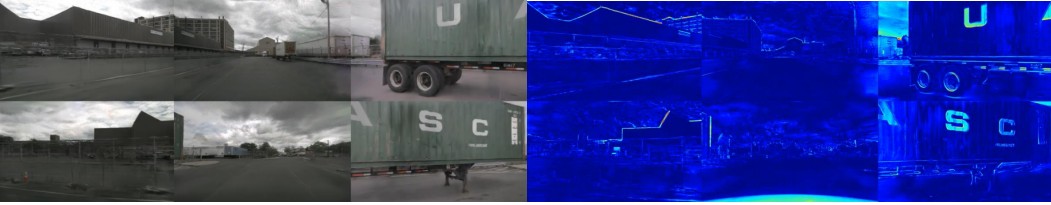

Baseline+BG

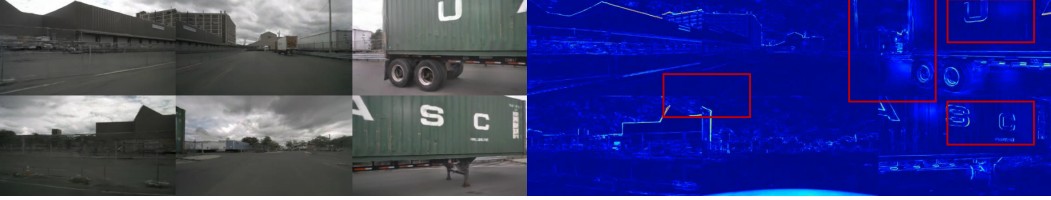

Baseline+Ours

Figure A1: **Qualitative Comparison of Photometric Correction with Baseline Methods.** These figures visualize the output of our method ('Baseline+Ours') against the ground truth, a baseline with appearance codes ('Baseline+AP'), and a baseline with a single bilateral grid ('Baseline+BG'). The accompanying error maps (blue indicating lower error, red higher) and highlighted red boxes demonstrate our method's superior ability to handle complex illumination and reduce artifacts compared to traditional approaches.

## C.2   Integration Evaluation with SOTA Methods

Tab. A9 and A10 provide comparisons with SOTA methods like ChatSim [4] and StreetGS [7] on the NuScenes dataset.

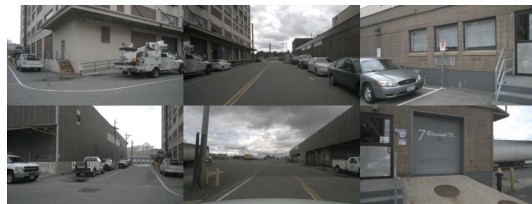

Ground Truth

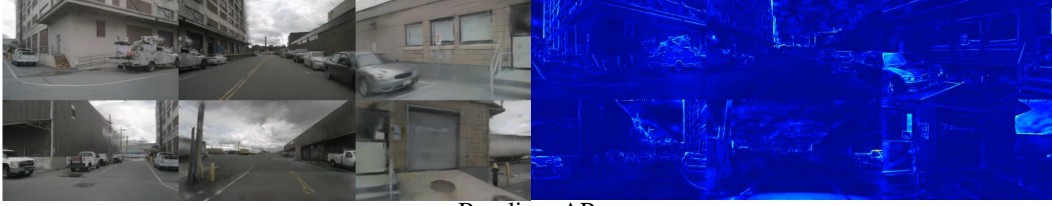

Baseline+AP

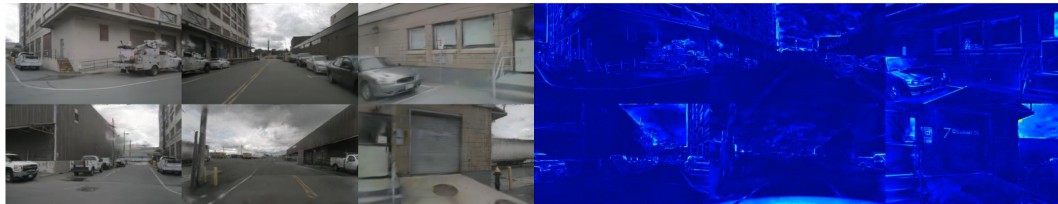

Baseline+BG

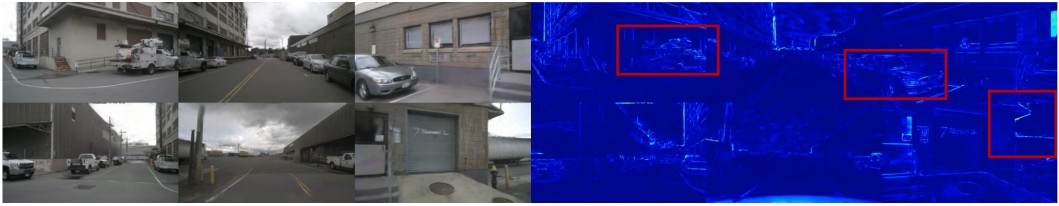

Baseline+Ours

Figure A2: **Qualitative Comparison of Photometric Correction with Baseline Methods.**

| Scene | Geometry Evaluation | | |
|---|---|---|---|
| | CD↓ | RMSE↓ | Depth↓ |
| 152 | 1.227 | 3.463 | 0.042 |
| 164 | 0.889 | 2.469 | 0.103 |
| 171 | 1.042 | 2.813 | 0.075 |
| 200 | 1.300 | 3.617 | 0.020 |
| 209 | 1.294 | 3.524 | 0.065 |
| 359 | 1.238 | 3.542 | 0.036 |
| 529 | 0.678 | 2.675 | 0.021 |
| 916 | 1.623 | 4.617 | 0.109 |
| Average | 1.161 | 3.340 | 0.059 |

Table A1: Detailed Scene-by-Scene Geometry Evaluation on the NuScenes [1]. This table presents key geometry metrics—Chamfer Distance (CD), Root Mean Square Error (RMSE), and Depth error—for individual scenes and averaged across the evaluated sequences from the NuScenes dataset.

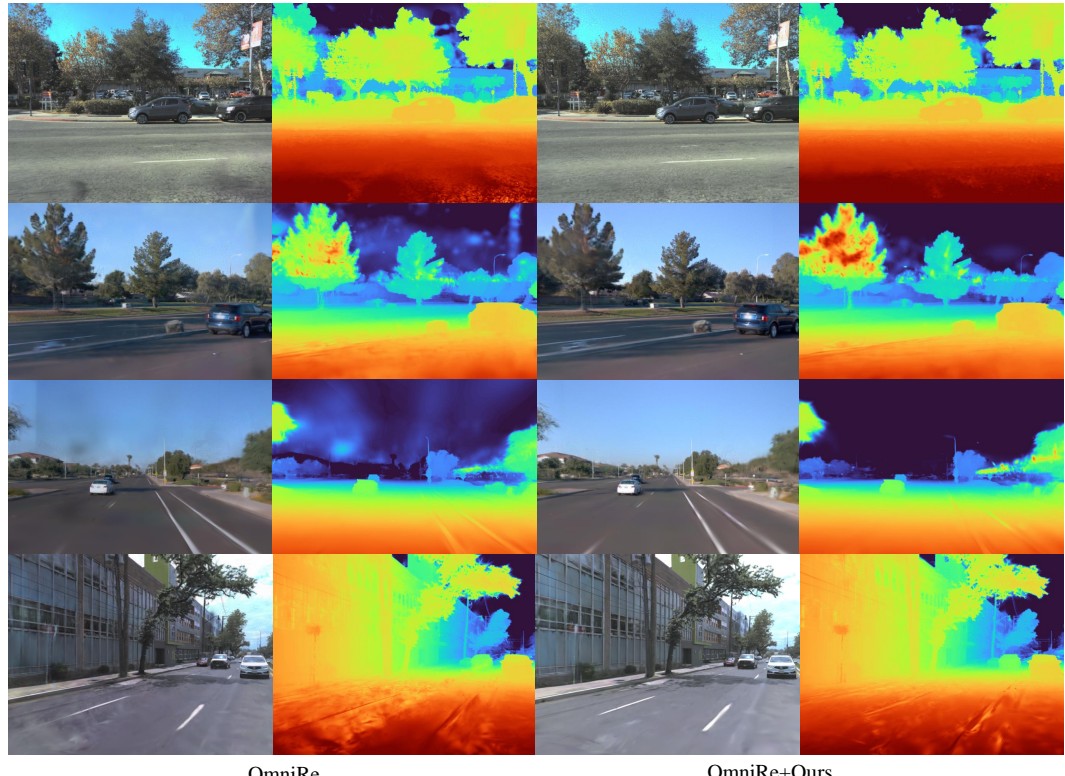

OmniRe       OmniRe+Ours

Figure A3: **Qualitative Comparison with the OmniRe.** This figure showcases a side-by-side visual comparison of renderings and depth maps produced by the original OmniRe framework versus OmniRe integrated with our proposed method ('OmniRe+Ours'). The results highlight the enhancements in image fidelity and depth map coherence achieved by our approach.

| Scene | Geometry Evaluation | | |
|---|---|---|---|
| | CD↓ | RMSE↓ | Depth↓ |
| 0 | 1.638 | 3.217 | 1.733 |
| 3 | 1.427 | 2.905 | 0.350 |
| 31 | 0.278 | 1.636 | 0.026 |
| 233 | 1.149 | 3.465 | 0.594 |
| 551 | 1.339 | 3.362 | 0.095 |
| 621 | 0.103 | 1.879 | 0.007 |
| Average | 0.989 | 2.744 | 0.467 |

Table A2: Detailed Scene-by-Scene Geometry Evaluation on the Waymo Open Dataset [2]. This table showcases the Chamfer Distance (CD), Root Mean Square Error (RMSE), and Depth error for specific scenes and their average, evaluating the geometric reconstruction accuracy on the Waymo Open Dataset.

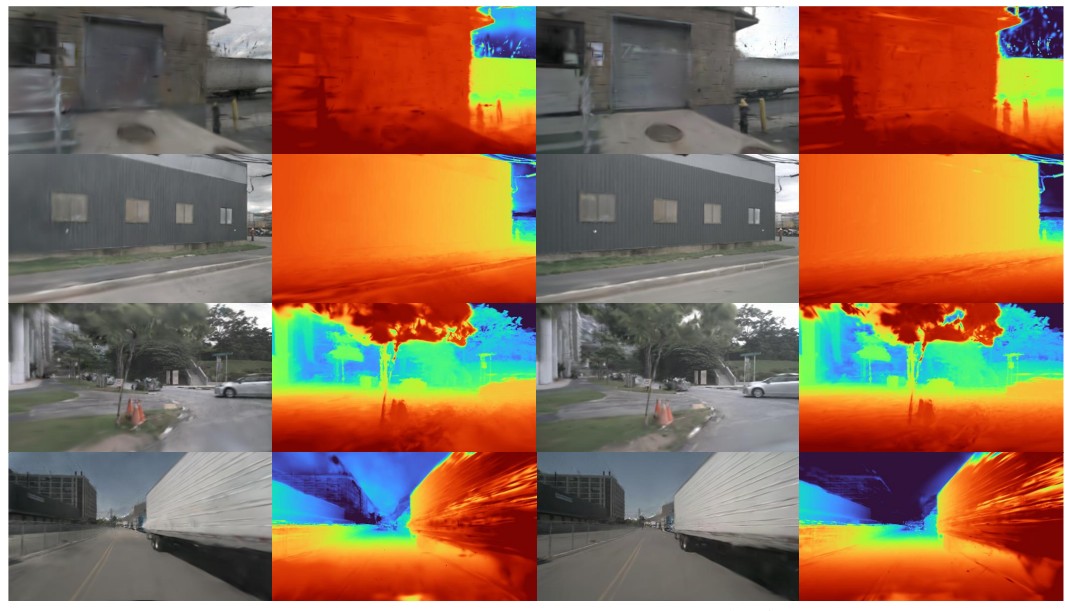

ChatSim                                    ChatSim+Ours

Figure A4: **Qualitative Comparison with the ChatSim.** Visual results comparing the ChatSim framework with ChatSim augmented by our method ('ChatSim+Ours'). The rendered images and corresponding depth maps illustrate the improvements in visual quality and geometric detail provided by our multi-scale bilateral grid framework.

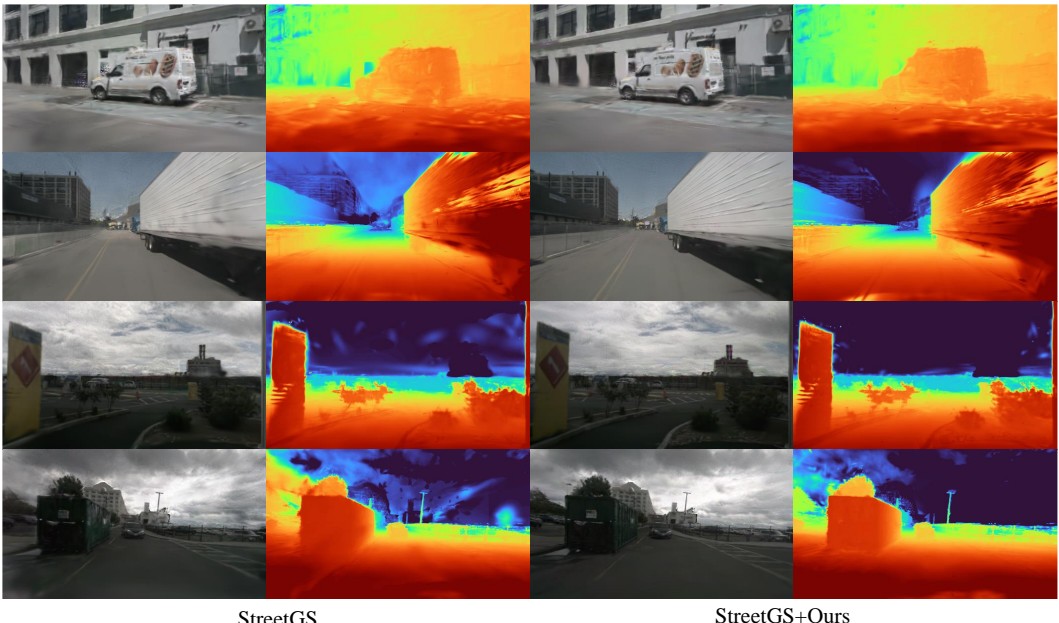

StreetGS                                    StreetGS+Ours

Figure A5: **Qualitative Comparison with the StreetGS.** This figure presents a visual comparison between the StreetGS framework and StreetGS enhanced with our method ('StreetGS+Ours'). The displayed images and depth maps demonstrate the capability of our approach to improve rendering realism and depth accuracy in challenging driving scenarios.

| Scene | Geometry Evaluation | | |
|---|---|---|---|
| | CD↓ | RMSE↓ | Depth↓ |
| 63 | 0.760 | 3.812 | 0.015 |
| 66 | 0.517 | 2.657 | 0.008 |
| 70 | 0.310 | 2.244 | 0.027 |
| 73 | 0.280 | 2.012 | 0.007 |
| 74 | 0.680 | 3.240 | 0.009 |
| 77 | 0.190 | 2.926 | 0.004 |
| 78 | 0.403 | 2.819 | 0.008 |
| 79 | 0.198 | 2.081 | 0.004 |
| 88 | 0.968 | 4.674 | 0.039 |
| 149 | 0.228 | 2.052 | 0.007 |
| Average | 0.453 | 2.852 | 0.013 |

Table A3: Detailed Scene-by-Scene Geometry Evaluation on the PandaSet [6]. This table provides a per-scene breakdown and average of Chamfer Distance (CD), Root Mean Square Error (RMSE), and Depth error, assessing geometric reconstruction performance on challenging nighttime scenarios from PandaSet.

| Scene | Geometry Evaluation | | |
|---|---|---|---|
| | CD↓ | RMSE↓ | Depth↓ |
| 0 | 0.522 | 3.001 | 0.023 |
| 1 | 2.071 | 5.845 | 0.085 |
| 2 | 0.461 | 3.226 | 0.011 |
| 3 | 0.348 | 3.677 | 0.014 |
| 4 | 1.198 | 4.485 | 0.059 |
| 5 | 0.694 | 5.069 | 0.040 |
| 6 | 0.967 | 6.193 | 0.075 |
| 8 | 0.743 | 4.436 | 0.045 |
| 9 | 0.257 | 1.952 | 0.005 |
| Average | 0.807 | 4.209 | 0.040 |

Table A4: Detailed Scene-by-Scene Geometry Evaluation on the Argoverse2 Dataset [5]. This table displays Chamfer Distance (CD), Root Mean Square Error (RMSE), and Depth error for individual sequences and their average, evaluating geometric accuracy on the Argoverse2 dataset.

| Scene | Scene Reconstruction | | | | | | | Novel View Synthesis | | | | | | |
|---|---|---|---|---|---|---|---|---|---|---|---|---|---|---|
| | Full Image | | | human | | vehicle | | Full Image | | | human | | vehicle | |
| | PSNR↑ | SSIM↑ | LPIPS↓ | PSNR↑ | SSIM↑ | PSNR↑ | SSIM↑ | PSNR↑ | SSIM↑ | LPIPS↓ | PSNR↑ | SSIM↑ | PSNR↑ | SSIM↑ |
| 152 | 27.39 | 0.839 | 0.204 | 29.33 | 0.873 | 27.76 | 0.862 | 23.57 | 0.695 | 0.234 | 25.04 | 0.655 | 22.07 | 0.579 |
| 164 | 26.52 | 0.829 | 0.232 | 24.75 | 0.789 | 25.60 | 0.793 | 23.16 | 0.711 | 0.262 | 19.86 | 0.464 | 21.74 | 0.595 |
| 171 | 26.97 | 0.833 | 0.249 | 24.05 | 0.652 | 26.22 | 0.812 | 24.39 | 0.748 | 0.273 | 22.28 | 0.568 | 22.51 | 0.639 |
| 200 | 27.80 | 0.835 | 0.193 | N/A | N/A | 28.08 | 0.832 | 25.28 | 0.745 | 0.211 | N/A | N/A | 24.11 | 0.643 |
| 209 | 29.07 | 0.866 | 0.185 | 28.20 | 0.802 | 28.84 | 0.836 | 26.18 | 0.785 | 0.204 | 25.42 | 0.694 | 25.07 | 0.683 |
| 359 | 27.95 | 0.859 | 0.168 | 26.98 | 0.795 | 27.25 | 0.839 | 24.43 | 0.723 | 0.190 | 24.06 | 0.629 | 22.72 | 0.647 |
| 529 | 29.90 | 0.893 | 0.133 | 24.25 | 0.736 | 27.80 | 0.856 | 27.40 | 0.826 | 0.146 | 24.18 | 0.712 | 24.84 | 0.745 |
| 916 | 25.88 | 0.819 | 0.181 | 28.16 | 0.792 | 26.95 | 0.839 | 22.73 | 0.678 | 0.208 | 25.44 | 0.637 | 22.75 | 0.634 |
| Average | 27.69 | 0.847 | 0.193 | 26.53 | 0.777 | 27.31 | 0.834 | 24.64 | 0.739 | 0.216 | 23.75 | 0.623 | 23.23 | 0.646 |

Table A5: Detailed Appearance Evaluation for Scene Reconstruction and Novel View Synthesis on the NuScenes Dataset [1]. This table presents PSNR, SSIM, and LPIPS metrics for the full image, as well as for 'human' and 'vehicle' classes, for both scene reconstruction and novel view synthesis tasks on various NuScenes sequences.

| Scene | Scene Reconstruction | | | | | | | Novel View Synthesis | | | | | | |
|---|---|---|---|---|---|---|---|---|---|---|---|---|---|---|
| | Full Image | | | human | | vehicle | | Full Image | | | human | | vehicle | |
| | PSNR↑ | SSIM↑ | LPIPS↓ | PSNR↑ | SSIM↑ | PSNR↑ | SSIM↑ | PSNR↑ | SSIM↑ | LPIPS↓ | PSNR↑ | SSIM↑ | PSNR↑ | SSIM↑ |
| 0 | 29.26 | 0.840 | 0.311 | N/A | N/A | 23.33 | 0.640 | 25.94 | 0.770 | 0.341 | N/A | N/A | 22.29 | 0.531 |
| 3 | 27.94 | 0.841 | 0.241 | 21.82 | 0.632 | N/A | N/A | 24.30 | 0.737 | 0.269 | 20.45 | 0.505 | N/A | N/A |
| 31 | 27.65 | 0.849 | 0.224 | 32.93 | 0.784 | 22.54 | 0.683 | 23.98 | 0.723 | 0.250 | 30.48 | 0.657 | 19.05 | 0.421 |
| 233 | 33.71 | 0.801 | 0.482 | N/A | N/A | 23.55 | 0.686 | 32.81 | 0.777 | 0.488 | N/A | N/A | 22.37 | 0.645 |
| 551 | 24.89 | 0.748 | 0.409 | 20.77 | 0.470 | 21.66 | 0.683 | 22.54 | 0.670 | 0.437 | 19.99 | 0.407 | 18.76 | 0.471 |
| 621 | 31.91 | 0.939 | 0.067 | 26.68 | 0.846 | 26.40 | 0.845 | 29.73 | 0.895 | 0.078 | 24.94 | 0.789 | 24.35 | 0.765 |
| Average | 29.23 | 0.836 | 0.289 | 25.55 | 0.683 | 23.50 | 0.707 | 26.55 | 0.762 | 0.310 | 23.97 | 0.590 | 21.36 | 0.567 |

Table A6: Detailed Appearance Evaluation for Scene Reconstruction and Novel View Synthesis on the Waymo Open Dataset [2]. This table details PSNR, SSIM, and LPIPS metrics for full images and specific object classes ('human', 'vehicle') during scene reconstruction and novel view synthesis on selected Waymo sequences.

| Scene | Scene Reconstruction | | | | | | | Novel View Synthesis | | | | | | |
|---|---|---|---|---|---|---|---|---|---|---|---|---|---|---|
| | Full Image | | | human | | vehicle | | Full Image | | | human | | vehicle | |
| | PSNR↑ | SSIM↑ | LPIPS↓ | PSNR↑ | SSIM↑ | PSNR↑ | SSIM↑ | PSNR↑ | SSIM↑ | LPIPS↓ | PSNR↑ | SSIM↑ | PSNR↑ | SSIM↑ |
| 63 | 29.95 | 0.905 | 0.226 | 29.44 | 0.783 | 22.32 | 0.726 | 27.32 | 0.854 | 0.258 | 26.93 | 0.613 | 18.49 | 0.447 |
| 66 | 31.73 | 0.931 | 0.198 | 29.94 | 0.845 | 20.78 | 0.681 | 28.92 | 0.890 | 0.215 | 26.84 | 0.738 | 19.55 | 0.591 |
| 70 | 31.09 | 0.909 | 0.260 | N/A | N/A | 21.35 | 0.740 | 28.35 | 0.862 | 0.287 | N/A | N/A | 21.20 | 0.674 |
| 73 | 32.13 | 0.931 | 0.198 | 31.26 | 0.868 | 23.33 | 0.806 | 29.16 | 0.889 | 0.214 | N/A | N/A | 21.84 | 0.706 |
| 74 | 29.85 | 0.914 | 0.207 | 30.99 | 0.874 | 24.44 | 0.849 | 27.13 | 0.862 | 0.228 | 27.82 | 0.740 | 22.23 | 0.740 |
| 77 | 32.60 | 0.937 | 0.165 | 32.28 | 0.852 | 25.42 | 0.855 | 29.77 | 0.895 | 0.178 | 29.02 | 0.730 | 22.12 | 0.721 |
| 78 | 31.30 | 0.922 | 0.181 | 33.54 | 0.896 | 25.27 | 0.812 | 28.39 | 0.869 | 0.197 | 30.74 | 0.809 | 22.86 | 0.683 |
| 79 | 31.42 | 0.902 | 0.228 | 31.91 | 0.850 | 26.06 | 0.843 | 28.14 | 0.832 | 0.251 | 28.98 | 0.726 | 23.08 | 0.702 |
| 88 | 25.09 | 0.787 | 0.269 | 22.10 | 0.670 | 21.90 | 0.738 | 21.95 | 0.623 | 0.305 | 20.19 | 0.362 | 18.46 | 0.444 |
| 149 | 32.31 | 0.926 | 0.202 | N/A | N/A | 24.04 | 0.791 | 29.73 | 0.891 | 0.215 | N/A | N/A | 23.37 | 0.728 |
| Average | 30.75 | 0.906 | 0.213 | 30.18 | 0.830 | 23.49 | 0.784 | 27.89 | 0.847 | 0.235 | 27.22 | 0.674 | 21.32 | 0.644 |

Table A7: Detailed Appearance Evaluation for Scene Reconstruction and Novel View Synthesis on PandaSet [6]. This table outlines PSNR, SSIM, and LPIPS metrics for full image and object-specific ('human', 'vehicle') evaluations in both scene reconstruction and novel view synthesis tasks using PandaSet nighttime sequences.

| Scene | Scene Reconstruction | | | | | | | Novel View Synthesis | | | | | | |
|---|---|---|---|---|---|---|---|---|---|---|---|---|---|---|
| | Full Image | | | human | | vehicle | | Full Image | | | human | | vehicle | |
| | PSNR↑ | SSIM↑ | LPIPS↓ | PSNR↑ | SSIM↑ | PSNR↑ | SSIM↑ | PSNR↑ | SSIM↑ | LPIPS↓ | PSNR↑ | SSIM↑ | PSNR↑ | SSIM↑ |
| 0 | 25.72 | 0.871 | 0.181 | 25.54 | 0.822 | 26.58 | 0.825 | 22.83 | 0.765 | 0.199 | 23.73 | 0.734 | 23.31 | 0.682 |
| 1 | 22.18 | 0.769 | 0.302 | 21.76 | 0.681 | 22.87 | 0.798 | 20.20 | 0.655 | 0.323 | 18.22 | 0.472 | 18.48 | 0.478 |
| 2 | 27.36 | 0.888 | 0.166 | 25.33 | 0.785 | 28.93 | 0.884 | 24.83 | 0.803 | 0.183 | 22.32 | 0.620 | 23.63 | 0.669 |
| 3 | 26.55 | 0.893 | 0.123 | 23.24 | 0.811 | 25.58 | 0.875 | 24.69 | 0.826 | 0.141 | 21.99 | 0.756 | 23.10 | 0.769 |
| 4 | 22.65 | 0.804 | 0.272 | 22.73 | 0.720 | 23.76 | 0.812 | 20.55 | 0.685 | 0.290 | 19.95 | 0.518 | 19.58 | 0.493 |
| 5 | 24.19 | 0.848 | 0.207 | 23.69 | 0.760 | 23.75 | 0.790 | 22.07 | 0.738 | 0.226 | 21.12 | 0.598 | 20.32 | 0.580 |
| 6 | 23.56 | 0.795 | 0.275 | 19.80 | 0.588 | 21.05 | 0.757 | 21.86 | 0.703 | 0.292 | 18.15 | 0.446 | 17.76 | 0.498 |
| 8 | 23.47 | 0.850 | 0.183 | 25.63 | 0.787 | 23.66 | 0.851 | 21.25 | 0.756 | 0.199 | 20.57 | 0.531 | 19.58 | 0.628 |
| 9 | 26.48 | 0.925 | 0.091 | 22.44 | 0.700 | 24.42 | 0.853 | 24.91 | 0.875 | 0.100 | 21.04 | 0.614 | 21.96 | 0.732 |
| Average | 24.68 | 0.849 | 0.200 | 23.35 | 0.739 | 24.51 | 0.827 | 22.58 | 0.756 | 0.217 | 20.79 | 0.588 | 20.86 | 0.615 |

Table A8: Detailed Appearance Evaluation for Scene Reconstruction and Novel View Synthesis on the Argoverse2 Dataset [5]. This table shows PSNR, SSIM, and LPIPS for full image and object-focused ('human', 'vehicle') assessments across scene reconstruction and novel view synthesis on the Argoverse2 dataset.

| Scene | Method | Reconstruction | | | Novel View Synthesis | | | Geometry | | |
|---|---|---|---|---|---|---|---|---|---|---|
| | | PSNR ↑ | SSIM ↑ | LPIPS ↓ | PSNR ↑ | SSIM ↑ | LPIPS ↓ | CD ↓ | RMSE ↓ | Depth ↓ |
| 152 | ChatSim | 25.37 | 0.811 | 0.251 | 1.592 | 3.637 | 0.085 | 22.72 | 0.694 | 0.276 |
| | Ours | 27.09 | 0.825 | 0.231 | 1.309 | 3.540 | 0.040 | 23.80 | 0.703 | 0.256 |
| 164 | ChatSim | 24.08 | 0.788 | 0.299 | 1.198 | 2.615 | 0.186 | 22.07 | 0.708 | 0.321 |
| | Ours | 26.16 | 0.802 | 0.273 | 0.972 | 2.523 | 0.101 | 23.79 | 0.718 | 0.293 |
| 171 | ChatSim | 25.00 | 0.801 | 0.310 | 1.420 | 2.981 | 0.105 | 23.31 | 0.736 | 0.326 |
| | Ours | 26.95 | 0.813 | 0.289 | 1.114 | 2.895 | 0.057 | 24.77 | 0.744 | 0.307 |
| 200 | ChatSim | 24.85 | 0.799 | 0.230 | 1.690 | 3.740 | 0.035 | 23.47 | 0.732 | 0.246 |
| | Ours | 27.54 | 0.816 | 0.220 | 1.289 | 3.619 | 0.017 | 25.43 | 0.745 | 0.235 |
| 209 | ChatSim | 25.77 | 0.818 | 0.268 | 1.841 | 3.805 | 0.118 | 24.30 | 0.763 | 0.281 |
| | Ours | 27.93 | 0.833 | 0.243 | 1.445 | 3.691 | 0.061 | 25.93 | 0.776 | 0.255 |
| 359 | ChatSim | 25.76 | 0.823 | 0.220 | 1.498 | 3.701 | 0.073 | 23.46 | 0.714 | 0.238 |
| | Ours | 27.17 | 0.834 | 0.208 | 1.317 | 3.647 | 0.036 | 24.31 | 0.719 | 0.227 |
| 529 | ChatSim | 27.19 | 0.864 | 0.191 | 1.071 | 2.853 | 0.077 | 25.70 | 0.812 | 0.201 |
| | Ours | 29.60 | 0.878 | 0.159 | 0.692 | 2.749 | 0.018 | 27.35 | 0.822 | 0.169 |
| 916 | ChatSim | 22.81 | 0.742 | 0.249 | 2.145 | 4.745 | 0.173 | 21.19 | 0.641 | 0.271 |
| | Ours | 23.95 | 0.759 | 0.232 | 1.750 | 4.635 | 0.102 | 21.96 | 0.655 | 0.253 |

Table A9: Comparative Performance Analysis against ChatSim on the NuScenes Dataset. This table provides a scene-by-scene comparison of our method ('Ours') with the ChatSim baseline across scene reconstruction (PSNR, SSIM, LPIPS), novel view synthesis (PSNR, SSIM, LPIPS), and geometry (CD, RMSE, Depth) metrics.

| Scene | Method | Reconstruction | | | Novel View Synthesis | | | Geometry | | |
|---|---|---|---|---|---|---|---|---|---|---|
| | | PSNR ↑ | SSIM ↑ | LPIPS ↓ | PSNR ↑ | SSIM ↑ | LPIPS ↓ | CD ↓ | RMSE ↓ | Depth ↓ |
| 152 | StreetGS | 25.49 | 0.813 | 0.254 | 1.611 | 3.648 | 0.089 | 22.73 | 0.695 | 0.280 |
| | Ours | 27.34 | 0.828 | 0.231 | 1.343 | 3.560 | 0.038 | 23.90 | 0.706 | 0.256 |
| 164 | StreetGS | 25.04 | 0.813 | 0.273 | 1.187 | 2.605 | 0.179 | 22.55 | 0.724 | 0.296 |
| | Ours | 27.41 | 0.827 | 0.248 | 0.956 | 2.523 | 0.096 | 24.29 | 0.734 | 0.269 |
| 171 | StreetGS | 25.62 | 0.811 | 0.305 | 1.355 | 2.958 | 0.115 | 23.63 | 0.743 | 0.323 |
| | Ours | 27.79 | 0.822 | 0.281 | 1.184 | 2.910 | 0.067 | 25.21 | 0.751 | 0.300 |
| 200 | StreetGS | 25.32 | 0.811 | 0.230 | 1.878 | 3.925 | 0.041 | 23.75 | 0.739 | 0.247 |
| | Ours | 28.12 | 0.829 | 0.219 | 1.418 | 3.817 | 0.020 | 25.67 | 0.752 | 0.235 |
| 209 | StreetGS | 26.53 | 0.840 | 0.243 | 1.932 | 3.813 | 0.125 | 24.77 | 0.779 | 0.258 |
| | Ours | 29.05 | 0.854 | 0.217 | 1.481 | 3.697 | 0.061 | 26.53 | 0.791 | 0.231 |
| 359 | StreetGS | 26.08 | 0.831 | 0.211 | 1.539 | 3.710 | 0.075 | 23.53 | 0.716 | 0.230 |
| | Ours | 27.63 | 0.843 | 0.199 | 1.299 | 3.659 | 0.039 | 24.45 | 0.723 | 0.218 |
| 529 | StreetGS | 27.50 | 0.871 | 0.186 | 1.065 | 2.883 | 0.063 | 25.96 | 0.819 | 0.197 |
| | Ours | 30.09 | 0.884 | 0.158 | 0.718 | 2.795 | 0.020 | 27.69 | 0.827 | 0.169 |
| 916 | StreetGS | 24.34 | 0.787 | 0.223 | 2.268 | 4.813 | 0.175 | 22.21 | 0.680 | 0.245 |
| | Ours | 25.80 | 0.804 | 0.204 | 1.781 | 4.709 | 0.106 | 23.11 | 0.692 | 0.226 |

Table A10: Comparative Performance Analysis against StreetGS on the NuScenes Dataset. This table presents a detailed per-scene comparison of our approach ('Ours') with the StreetGS baseline, evaluating scene reconstruction (PSNR, SSIM, LPIPS), novel view synthesis (PSNR, SSIM, LPIPS), and geometry (CD, RMSE, Depth) metrics.

Table A11: Detailed comparison on **Challenging Argoverse Scenarios** (6 scenes).

| Method | Reconstruction | | | | | Novel View Synthesis | |
|---|---|---|---|---|---|---|---|
| | PSNR ↑ | SSIM ↑ | CD ↓ | RMSE ↓ | Depth ↓ | PSNR ↑ | SSIM ↑ |
| OmniRe (Baseline) | 23.95 | 0.850 | 0.964 | 3.647 | 0.098 | 22.10 | 0.761 |
| OmniRe w/ AC | 23.97 | 0.848 | 0.923 | 3.642 | 0.097 | 22.09 | 0.759 |
| OmniRe w/ BG | 24.29 | 0.855 | 0.807 | 3.586 | 0.047 | 22.32 | 0.768 |
| **Ours** | **25.29** | **0.863** | **0.726** | **3.564** | **0.027** | **22.96** | **0.773** |

Table A12: Detailed comparison on **Challenging Waymo Scenarios** (6 scenes).

| Method | Reconstruction | | | | | Novel View Synthesis | |
|---|---|---|---|---|---|---|---|
| | PSNR ↑ | SSIM ↑ | CD ↓ | RMSE ↓ | Depth ↓ | PSNR ↑ | SSIM ↑ |
| OmniRe (Baseline) | 29.28 | 0.833 | 0.352 | 1.917 | 0.061 | 27.37 | 0.791 |
| OmniRe w/ AC | 29.31 | 0.833 | 0.343 | 1.915 | 0.053 | 27.30 | 0.789 |
| OmniRe w/ BG | 29.61 | 0.836 | 0.333 | 1.893 | 0.042 | 27.56 | 0.792 |
| **Ours** | **31.23** | **0.841** | **0.272** | **1.846** | **0.021** | **28.34** | **0.793** |

Table A13: Detailed comparison on **Challenging Pandaset Scenarios** (6 scenes).

| Method | Reconstruction | | | | | Novel View Synthesis | |
|---|---|---|---|---|---|---|---|
| | PSNR ↑ | SSIM ↑ | CD ↓ | RMSE ↓ | Depth ↓ | PSNR ↑ | SSIM ↑ |
| OmniRe (Baseline) | 27.28 | 0.859 | 0.634 | 3.963 | 0.029 | 23.91 | 0.721 |
| **Ours** | **27.88** | **0.863** | **0.572** | **3.951** | **0.022** | **24.22** | **0.724** |

Table A14: Comparison results of **vehicle-specific metrics**, averaged across 20 scenes from all four datasets.

| Method | PSNR ↑ | SSIM ↑ | CD ↓ |
|---|---|---|---|
| OmniRe (Baseline) | 24.24 | 0.784 | 8.675 |
| **Ours** | **24.80** | **0.791** | **7.400** |

## C.3  Performance on Challenging Scenarios

To validate robustness, we curated subsets of challenging scenarios with extreme photometric inconsistencies (e.g., night scenes, sun glare, rain, and complex reflections). As shown in Tables A11, A12, and A13, our method's performance gap widens significantly compared to all baselines (OmniRe, w/ AC, w/ BG) in these difficult conditions, demonstrating the superior robustness of our multi-scale design.

- **Challenging Argoverse (6 scenes):** `49e970c4` (Extreme Lighting, Overexposure, Lens Flare), `8184872e` (Complex Shadows, Direct Sunlight), `91923e20` (Extreme Lighting, Camera Artifacts), `a8a2fbc2` (Low-Angle Sun), `b403f8a3` (Overpass Structure, Extreme Lighting Transitions), `eaaf5ad3` (Multiple Artificial Lights, Wet Surfaces, Rainy Conditions).
- **Challenging Waymo (6 scenes):** `segment-10` (Night Scene, Artificial Lighting, Lens Flare), `segment-30` (Night Scene, Rainy Conditions, Specular Reflections), `segment-539` (High Dynamic Range, Low-Angle Sun, Sun Glare), `segment-550` (High Dynamic Range, Storefront Reflections, Hard Shadows), `segment-561` (Low-Angle Sun, Lens Flare, Overexposure), `segment-570` (Rainy Conditions, Low Light, Motion Blur).
- **Challenging Pandaset (6 scenes):** 19, 21, 29, 48, 52 (all Low-Angle Sun with Hard/Deep Shadows or High Dynamic Range), 63 (Night Scene, Multiple Light Sources, Glare).

## C.4  Vehicle-Specific Metrics

To specifically evaluate performance on challenging, highly reflective surfaces such as car windows, we used semantic masks to isolate vehicles. Table A14 shows that our method improves not only appearance (PSNR) but also, crucially, geometry (CD) for these specific objects. This supports our claim that our method mitigates the negative geometric impact of such severe view-dependent effects.

# D  Quantitative Analysis of Photometric Inconsistency

To rigorously validate the learned affine transformations discussed in the main paper, we compute the Kullback-Leibler (KL) Divergence between our model's learned distribution and a ground-truth (GT) distribution of real-world photometric inconsistency.

**Experimental Setup:** We construct the GT distribution $P(\Delta L)$ by sampling corresponding pixels representing the same 3D world points across different camera views in the nuScenes dataset, using ground-truth LiDAR points and camera parameters. The luminance difference $\Delta L$ for all corresponding pairs forms the GT histogram. We then measure the KL Divergence between this $P(\Delta L)$ and the distributions learned by our model and the baseline.

**Results:** As shown in Table A15, the distribution learned by our multi-scale method achieves a significantly lower KL Divergence (0.79) compared to the single-scale baseline (1.25). This quantitatively confirms that our model learns a distribution that more accurately reflects real-world photometric inconsistencies.

Table A15: Comparison of KL Divergence against the ground-truth (GT) photometric inconsistency distribution from the nuScenes dataset. Lower is better, indicating a more realistic learned distribution.

| Method | KL Divergence (vs. Dataset GT) ↓ |
|---|---|
| Single Bilateral Grids' Affine Transformation | 1.25 |
| **Multi-scale Bilateral Grids' Affine Transformation (Ours)** | **0.79** |

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
