# OpenReview forum: "Unifying Appearance Codes and Bilateral Grids for Driving Scene Gaussian Splatting"
_NeurIPS.cc/2025/Conference — NeurIPS 2025 poster_

### Official Review · Reviewer_dvs1 · 2025-06-04

**Clarity:** 3
**Significance:** 3
**Originality:** 3
**Rating:** 4
**Confidence:** 4

**Summary:**

This paper proposes a novel multi-scale bilateral grid that unifies appearance codes and bilateral grids to handle the photometric inconsistency problem in reconstructing autonomous driving scenes. The multi-scale bilateral grid provides affine color transformation matrices for pixels in a coarse-to-fine manner, thereby achieving global and local photometric corrections. The experiments demonstrate the effectiveness of proposed methods.

**Questions:**

+ Can the multi-scale bilateral grid handle the reflection problem of several car window glass for better geometric accuracy?
+ The improvements shown in Table 1 on the Waymo and Argoverse datasets are not obvious compared to the "Baseline" which has no photometric corrections.
+ The efficiency analysis in Table 4 probably requires a "Baseline"(without any photometric corrections) model to further show the computational overhead of the proposed methods.

**Ethical Concerns:**

["NO or VERY MINOR ethics concerns only"]

**Final Justification:**

My main concerns about the performance of the proposed methods have been addressed. In more challenging scenes provided by the authors, the proposed methods obtain an obvious improvement in comparison to the traditional appearance codings. However, just as Reviewer 2hvk said, the proposed appearance coding might be a smart engineering combination of known components. In general, I think it is a simple but effective work. Therefore, I tend to keep my rating as boardline accept.

**Limitations:**

Yes, the limitation has been mentioned.

**Paper Formatting Concerns:**

No formatting issue.

**Quality:**

3

**Strengths And Weaknesses:**

**Strengths**
+ The paper is well written and clearly expressed.
+ The proposed multi-scale bilateral grid seems novel and effective.
+ The experiments are thorough. The authors evaluate their methods on different datasets and baselines, and the improvements shown in several experiments are obvious.

**Weakness**
+ The improvements shown in Table 1 on the Waymo and Argoverse datasets are not obvious compared to the "Baseline" which has no photometric corrections.
+ The efficiency analysis in Table 4 probably requires a "Baseline"(without any photometric corrections) model to further show the computational overhead of the proposed methods.

---

> ### Author Rebuttal · Authors · 2025-07-31
>
> We thank reviewer **dvs1** for the constructive review and for recognizing the novelty and thoroughness of our experiments.
>
> ---
>
> #### **W1 / Q2: The improvements shown in Table 1 on the Waymo and Argoverse datasets are not obvious compared to the "Baseline" which has no photometric corrections.**
>
> - The core reasons for this are **a.** **dataset characteristics** and **b. intrinsic capabilities of gaussian splatting**.
> - We would like to clarify that our method still provides a significant improvement in **geometric accuracy** under these reason. Meanwhile, we provide additional evaluation.
>
>     - **Dataset Characteristics.** In our evaluation, the selected test sequences from the Argoverse and PandaSet datasets feature relatively more consistent lighting conditions. And our method mainly focuses on appearance inconsistencies, making improvements not obvious.
>     - **Gaussian Splatting has capabilities of fitting appearance variance using "floater" [A].** Gaussian Splatting model has a native capacity to handle minor appearance variations.
>         1. It can model some inconsistencies by adjusting its **Spherical Harmonics (SH) coefficients** and by using **semi-transparent Gaussians**.
>         2. When these inconsistencies are less severe, a baseline model might "bake" these small errors into 3D geometry, which is why the PSNR/SSIM improvement of our method is more modest on those datasets.
>         3. However, it would harm geometry fidelity, as proved in **Table 1,** geometry metrics of baseline obviously worse.
>     - Our method prevents this, leading to consistently better geometric outcomes across all tested datasets, which is the core contribution of our paper.
> - **Additional experiments to validate.** To further validate our method's performance, we conduct additional experiments on a curated subset of scenes featuring more extreme lighting and weather conditions. The scenes are listed here:
>
> > Table R10: Additional challenging Argoverse scenarios.
> |scene ID|description |
> | --- | --- |
> | 49e970c4 | Extreme Lighting, Overexposure, Lens Flare |
> | 8184872e | Complex Shadows, Direct Sunlight, Information Loss |
> | 91923e20 | Extreme Lighting, Deep Shadows, Camera Artifacts. |
> | a8a2fbc2 | Low-Angle Sun, Complex Shadows, Texture Disentanglement. |
> | b403f8a3 | Overpass Structure, Extreme Lighting Transitions, Complex Indirect Illumination |
> | eaaf5ad3 | Multiple Artificial Lights, Wet Surfaces, Rainy Conditions |
>
> > Table R11: Evaluation results of our multi-scale bilateral grids on additional challenging Argoverse scenarios:
> |  | Recon. PSNR$\uparrow$ | Recon. SSIM$\uparrow$ | Recon. CD$\downarrow$ | Recon. RMSE$\downarrow$ | Recon. Depth$\downarrow$ | NVS PSNR$\uparrow$ | NVS SSIM$\uparrow$ |
> | --- | --- | --- | --- | --- | --- | --- | --- |
> | 49e970c4 | 23.35 | 0.789 | 1.229 | 5.175 | 0.060 | 21.33 | 0.673 |
> | 8184872e | 24.23 | 0.864 | 0.666 | 3.303 | 0.022 | 21.76 | 0.766 |
> | 91923e20 | 25.86 | 0.876 | 0.820 | 3.342 | 0.019 | 23.48 | 0.789 |
> | a8a2fbc2 | 25.13 | 0.823 | 0.774 | 3.513 | 0.033 | 22.65 | 0.716 |
> | b403f8a3 | 27.55 | 0.926 | 0.298 | 2.822 | 0.004 | 25.25 | 0.870 |
> | eaaf5ad3 | 25.60 | 0.899 | 0.566 | 3.228 | 0.022 | 23.28 | 0.824 |
> | Average | **25.29** | **0.863** | **0.726** | **3.564** | **0.027** | **22.96** | **0.773** |
>
> > Table R12: Evaluation results of OmniRe on additional challenging Argoverse scenarios:
> |  | Recon. PSNR$\uparrow$ | Recon. SSIM$\uparrow$ | Recon. CD$\downarrow$ | Recon. RMSE$\downarrow$ | Recon. Depth$\downarrow$ | NVS PSNR$\uparrow$ | NVS SSIM$\uparrow$ |
> | --- | --- | --- | --- | --- | --- | --- | --- |
> | 49e970c4 | 21.99 | 0.772 | 1.474 | 5.270 | 0.129 | 20.37 | 0.659 |
> | 8184872e | 22.79 | 0.850 | 0.862 | 3.389 | 0.101 | 20.79 | 0.751 |
> | 91923e20 | 24.47 | 0.859 | 1.171 | 3.463 | 0.124 | 22.67 | 0.776 |
> | a8a2fbc2 | 23.13 | 0.805 | 1.107 | 3.602 | 0.140 | 21.20 | 0.701 |
> | b403f8a3 | 26.65 | 0.922 | 0.396 | 2.862 | 0.014 | 24.77 | 0.869 |
> | eaaf5ad3 | 24.69 | 0.889 | 0.776 | 3.295 | 0.083 | 22.79 | 0.810 |
> | Average | 23.95 | 0.850 | 0.964 | 3.647 | 0.098 | 22.10 | 0.761 |
> - In these challenging Argoverse scenarios, our method outperforms OmniRe by an average of **1.338 dB** in PSNR.
>
> > Table R13: Additional challenging Waymo scenarios.
> |scene ID|description |
> | --- | --- |
> | 10 | Night Scene, Artificial Lighting, Lens Flare |
> | 30 | Night Scene, Rainy Conditions, Specular Reflections |
> | 539 | High Dynamic Range, Low-Angle Sun, Sun Glare |
> | 550 | High Dynamic Range, Storefront Reflections, Hard Shadows |
> | 561 | Low-Angle Sun, Lens Flare, Overexposure |
> | 570 | Rainy Conditions, Low Light, Motion Blur |
>
> > Table R14: Evaluation results of our multi-scale bilateral grids on additional challenging Waymo scenarios:
> |  | Recon. PSNR$\uparrow$ | Recon. SSIM$\uparrow$ | Recon. CD$\downarrow$ | Recon. RMSE$\downarrow$ | Recon. Depth$\downarrow$ | NVS PSNR$\uparrow$ | NVS SSIM$\uparrow$ |
> | --- | --- | --- | --- | --- | --- | --- | --- |
> | 10 | 31.46 | 0.716 | 0.155 | 1.167 | 0.007 | 30.10 | 0.674 |
> | 30 | 29.29 | 0.734 | 0.528 | 2.731 | 0.024 | 27.98 | 0.698 |
> | 539 | 28.17 | 0.826 | 0.385 | 2.530 | 0.052 | 24.60 | 0.746 |
> | 550 | 36.26 | 0.962 | 0.050 | 1.049 | 0.001 | 32.24 | 0.943 |
> | 561 | 32.68 | 0.944 | 0.094 | 1.894 | 0.002 | 29.51 | 0.898 |
> | 570 | 29.52 | 0.861 | 0.421 | 1.707 | 0.043 | 25.60 | 0.802 |
> | Average | **31.23** | **0.841** | **0.272** | **1.846** | **0.021** | **28.34** | **0.793** |
>
> > Table R15: Evaluation results of OmniRe on additional challenging Waymo scenarios.
> |  | Recon. PSNR$\uparrow$ | Recon. SSIM$\uparrow$ | Recon. CD$\downarrow$ | Recon. RMSE$\downarrow$ | Recon.  Depth$\downarrow$ | NVS PSNR$\uparrow$ | NVS SSIM$\uparrow$ |
> | --- | --- | --- | --- | --- | --- | --- | --- |
> | 10 | 30.80 | 0.711 | 0.197 | 1.189 | 0.020 | 29.79 | 0.672 |
> | 30 | 28.52 | 0.724 | 0.559 | 2.765 | 0.043 | 26.88 | 0.689 |
> | 539 | 25.01 | 0.822 | 0.617 | 2.704 | 0.146 | 21.86 | 0.739 |
> | 550 | 32.74 | 0.953 | 0.052 | 1.133 | 0.001 | 32.13 | 0.946 |
> | 561 | 31.12 | 0.942 | 0.102 | 1.898 | 0.003 | 28.57 | 0.898 |
> | 570 | 27.50 | 0.847 | 0.585 | 1.815 | 0.155 | 24.97 | 0.800 |
> | Average | 29.28 | 0.833 | 0.352 | 1.917 | 0.061 | 27.37 | 0.791 |
> - In these challenging Waymo scenarios, our method outperforms OmniRe by an average of **1.950 dB** in PSNR.
>
> ---
>
> #### **W2 / Q3: The efficiency analysis in Table 4 probably requires a "Baseline"(without any photometric corrections) model to further show the computational overhead of the proposed methods.**
>
> - We thank the reviewer for this constructive suggestion.
>     - We agree that including a "Baseline" model without photometric corrections will provide a clearer benchmark for the efficiency analysis. We will update Table 4 accordingly in the final version.
> - We wish to clarify that the computational overhead of the baseline is negligible and **nearly identical to our "Baseline w/AC" model**, that is the reason why we didn't add another line to show.
>
> ---
>
> #### **Q1: Can the multi-scale bilateral grid handle the reflection problem of several car window glass for better geometric accuracy?**
>
> - Our method can **effectively mitigate their negative impact** on geometric reconstruction.
> - It achieves this by treating reflections as a source of severe, view-dependent photometric inconsistency.
>     - When reflections cause a car window to appear bright from one angle and dark from another, our model learns a transformation to reduce this mismatch.
>     - By improving photometric consistency on these surfaces, our method prevents the optimization process from generating erroneous geometric artifacts, such as floaters, which leads to improved overall geometric accuracy.
> - We provide both qualitative and quantitative evidence to support this:
>     - **Qualitative Results:**
>         - Figure 3(a) in the main paper demonstrates our method’s ability to handle specular highlights, a phenomenon physically related to reflections.
>         - Figure A2 in the supplementary material provides a direct visual comparison, showing that our approach produces a better visual appearance and fewer artifacts in the reflective regions of a vehicle compared to baselines.
>     - **Quantitative Results:** To quantitatively evaluate our method on challenging surfaces, we focus on vehicle-specific metrics.
>         - We use semantic masks to isolate vehicles—which often feature reflective windows—and compute metrics for both appearance and geometric.
>         - The following Table presents the results, averaged across 20 scenes (5 scenes each from the Waymo, Pandaset, Argoverse, and NuScenes).
>         - The data demonstrate its effectiveness in mitigating the negative impact of reflective surfaces.
>
> > Table R16: Comparison results of vehicle-specific metrics.
> |  | PSNR$\uparrow$ | SSIM$\uparrow$ | CD$\downarrow$ |
> | --- | --- | --- | --- |
> | OmniRe | 24.24 | 0.784 | 8.675 |
> | Ours | **24.80** | **0.791** | **7.400** |
>
> ---
>
> We appreciate all your constructive comments and we will improve our work according to them in the near future. If the reviewer has any follow-up questions, we are happy to discuss them.
>
> > [A] Zhang, Dongbin, et al. "Gaussian in the wild: 3d gaussian splatting for unconstrained image collections." European Conference on Computer Vision. Cham: Springer Nature Switzerland, 2024.

---

> > ### Comment · Reviewer_dvs1 · 2025-08-03
> >
> > Thanks for your answers. I think most of my concerns have been addressed. However, I wonder if the authors could provide several results of "Baseline w/AC" on your additional challenging Argoverse/Waymo scenarios (in your rebuttal) to further demonstrate your performance.

---

> ### Author Response · Authors · 2025-08-04
>
> We sincerely appreciate your comments. We have completed the additional experiments as requested and present the results below.
>
> ---
>
> - We evaluated the results of "Baseline w/AC".
> - The results are averaged over the challenging subsets for Argoverse (Table R10) and Waymo (Table R13).
> - To provide a comprehensive analysis, we also provide the results of "Baseline w/BG", this aligns with Table 1, 2 in our main paper.
>
> > Table: Comparison results of additional challenging Argoverse scenarios.
> >
>
> |  | Recon. PSNR$\uparrow$ | Recon. SSIM$\uparrow$ | Recon. CD$\downarrow$ | Recon. RMSE$\downarrow$ | Recon. Depth$\downarrow$ | NVS PSNR$\uparrow$ | NVS SSIM$\uparrow$ |
> | --- | --- | --- | --- | --- | --- | --- | --- |
> | OmniRe w/o AC | 23.95 | 0.850 | 0.964 | 3.647 | 0.098 | 22.10 | 0.761 |
> | OmniRe w/ AC | 23.97 | 0.848 | 0.923 | 3.642 | 0.097 | 22.09 | 0.759 |
> | OmniRe w/ BG | 24.29 | 0.855 | 0.807 | 3.586 | 0.047 | 22.32 | 0.768 |
> | Ours | **25.29** | **0.863** | **0.726** | **3.564** | **0.027** | **22.96** | **0.773** |
>
> > Table: Comparison results of additional challenging Waymo scenarios.
> >
>
> |  | Recon. PSNR$\uparrow$ | Recon. SSIM$\uparrow$ | Recon. CD$\downarrow$ | Recon. RMSE$\downarrow$ | Recon. Depth$\downarrow$ | NVS PSNR$\uparrow$ | NVS SSIM$\uparrow$ |
> | --- | --- | --- | --- | --- | --- | --- | --- |
> | OmniRe w/o AC | 29.28 | 0.833 | 0.352 | 1.917 | 0.061 | 27.37 | 0.791 |
> | OmniRe w/ AC | 29.31 | 0.833 | 0.343 | 1.915 | 0.053 | 27.3 | 0.789 |
> | OmniRe w/ BG | 29.61 | 0.836 | 0.333 | 1.893 | 0.042 | 27.56 | 0.792 |
> | Ours | **31.23** | **0.841** | **0.272** | **1.846** | **0.021** | **28.34** | **0.793** |
>
> We also find:
>
> - On theses chanllenging scenarios.
>     - Single bilateral grids also present better improvement compare to normal scenerios
>     - Our method still achieves the best results
> - Our insight is that these challenging scenarios feature significant local variance.
>
> ---
>
> We will add this additional results and analysis to the final paper. Thank you again for your constructive guidance.

---

> > ### Comment · Reviewer_dvs1 · 2025-08-05
> >
> > Thanks for your responses. My concerns have been addressed, and I have no further questions.

---

### Official Review · Reviewer_2hvk · 2025-06-09

**Clarity:** 2
**Significance:** 2
**Originality:** 2
**Rating:** 3
**Confidence:** 4

**Summary:**

This paper tackles inconsistent appearance in 3DGS driving scenes with a multi-scale bilateral grid. It works hierarchically to provide both global (coarse levels) and local (fine levels) appearance correction. Experiments on four driving datasets show improved geometry and reduced visual artifacts, outperforming standard baselines like appearance codes.

**Questions:**

1. Please clarify the implementation details for the "Baseline" method used in comparisons (e.g., Fig. 3).
2. The paper evaluates dynamic scenes using static frames, which can be prone to cherry-picking favorable results. Could the authors provide video results for the driving sequences?
3. Autonomous driving scenarios are inherently dynamic. The paper mentions that modeling "extremely complex, fast-moving, or highly non-rigid objects" remains a challenge.  Could you provide some qualitative examples of these failure cases?
4. The ablation in Table 5 is concerning: adding finer grids can harm geometric performance. This challenges the robustness of the fixed grid architecture. How sensitive is performance to this specific three-level design? Does the optimal grid configuration change per-scene (e.g., night vs. day), requiring manual tuning?
5. The method performs appearance correction in 2D as a post-processing step. How does this compare conceptually to methods that disentangle appearance in 3D (e.g., Relightable 3DGS)? What are the fundamental limitations of a 2D approach for handling 3D view-dependent effects like specular highlights?

**Ethical Concerns:**

["NO or VERY MINOR ethics concerns only"]

**Final Justification:**

After considering the rebuttal and the discussion, I find that some of my initial concerns have been clarified, but the key issues are not fully resolved. I weighed the strengths of the work against the remaining weaknesses and concluded that my original score best reflects the overall quality and contribution of the paper.

**Limitations:**

Yes. The authors discuss limitations in the conclusion, primarily: increased computational overhead and challenges with modeling highly dynamic or non-rigid objects.

**Paper Formatting Concerns:**

No major formatting issues found.

**Quality:**

3

**Strengths And Weaknesses:**

**Strengths**:
- The paper tackles a critical and well-known challenge in real-world 3D reconstruction. Photometric inconsistency in driving data is a known source of artifacts and geometric errors for 3DGS. Solving this is valuable for downstream tasks like simulation and planning.
- The method shows strong quantitative and qualitative results. Evaluation is thorough, covering four large driving datasets (Waymo, etc.) and multiple baselines. The handling of dynamic objects further demonstrates practical relevance.


**Weaknesses**:
- The novelty is limited. The proposed multi-scale grid is a heuristic that combines two existing ideas (global and per-pixel correction), rather than a fundamental new technique. The presentation overstates the novelty of what is effectively a smart engineering combination of known components.
- The analysis in Section 4.3 is unconvincing. The central claim—that a "flatter" histogram signifies a better model—is an unproven assertion. A flat distribution could equally imply that the model is learning unstructured noise or is poorly constrained, whereas a peaky distribution might indicate an efficient model that has correctly identified the primary sources of error. The analysis shows a correlation between flatter histograms and better results, but fails to establish causation; Also Lacking a ground-truth comparison, it arbitrarily claims one learned distribution is superior to another, making the entire analysis feels hand-wavy and lacks rigor.

---

> ### Author Rebuttal · Authors · 2025-07-31
>
> Thank reviewer **2hvk** for detailed review and constructive feedback. We appreciate the opportunity to clarify our work.
>
> ---
>
> #### **W1: The novelty is limited.**
>
> - Our multi-scale bilateral grids architecture is a deliberate design choice, aimed at creating a **pragmatic and robust solution**, and we believe the specific way these components are unified represents a novel and valuable contribution.
>     - Our goal was to address the critical problem of geometric inaccuracy caused by photometric inconsistencies in autonomous driving scenes.
>     - After exploring various approaches, including more complex 3D disentanglement techniques, we focused on developing a method that strikes a crucial balance between reconstruction performance, computational efficiency, and ease of integration.
> - While we build upon known components, our contribution is a **principled unification** that resolves their opposing weaknesses.
>     - Our framework overcomes the optimization instability of fine-grained bilateral grids while adding the local modeling capability that global appearance codes lack.
>     - The core technical insight is the **coarse-to-fine residual refinement** process, where a stable global transformation is learned first, followed by residual corrections from finer grids.
> - The success of this approach is demonstrated by its strong and consistent performance.
>     - The **identical architecture and hyperparameters** were used across four diverse datasets (including day and night scenes) without any per-scene tuning, proving its robustness and generalizability.
>     - We believe this balance of performance, efficiency, and stability offers a valuable and practical solution that addresses a key gap for real-world driving scene reconstruction.
>
> ---
>
> #### **W2: The analysis in Section 4.3 is unconvincing.**
>
> - We appreciate reviewer **2hvk** for this sharp critique and acknowledge that our original explanation required further clarification.
> - We totally agree that a more rigorous analysis is necessary to elucidate why the multi-scale bilateral grids achieve superior performance and provide valuable practical guidance for future research
>
> To quantify photometric inconsistency between camera views, we design an experiment to make further analysis from the real-world driving dataset.
>
> - We regret the front camera view as reference, left-front and right-front camera as target views
> - Using ground truth LiDAR points $L_w$, camera poses $P_{ref},P_{trg}^i$, and intrinsic matrix $K_{ref},K_{trg}^i$, we get corresponding pixels that represent the same 3D world position among reference views and target views.
>     - For each corresponding pair, the color values are sampled from their respective images. These colors are converted to luminance, and the difference $\Delta L$ between them can be calculated.
>     - This process is repeated for all initial LiDAR points to create a large set of luminance differences. A histogram of this set provides the final statistical distribution, $P(\Delta L)$ which characterizes the inherent photometric inconsistency between the camera views.
>     - Compare $P(\Delta L) $ distribution of affine transformation using KL Divergence. We can further prove which distribution better models the real-world appearance inconsistency of the dataset.
> - The experiment on our NuScenes test set show that our method has more similar distribution with GT.
>
> > Table R10: Comparison of KL Divergence.
> |             | KL Divergence with dataset distribution |
> |---------|----------|
> | multi-scale bilateral grids' affine transformation | 0.79  |
> | single bilateral grids' affine transformation      | 1.25 |
>
> - Our insight is color inconsistency in real scenes is affected by random factors such as lighting, which requires more diverse modeling capabilities.
>
> ---
>
> #### **Q1: Please clarify the implementation details for the "Baseline" method used in comparisons (e.g., Fig. 3).**
>
> - The "**Baseline**" method used in our main comparison tables (**Table 1, 3, 4**) and figures (**Figure 2, 3, 4**) refers to **OmniRe [A]**. As stated in **Section 4.1**.
> - The variants presented in our main ablation study are:
>     - **Baseline**: OmniRe.
>     - **Baseline w/AC**: OmniRe with appearance codes added.
>     - **Baseline w/BG**: OmniRe with single bilateral grids added.
>     - **Ours**: OmniRe [5] with our proposed multi-scale bilateral grids.
> - We will revise to make these definitions explicit in the text and figure captions to avoid confusion. Thank reviewer **2hvk** for suggestions.
>
> ---
>
> #### **Q2: Could the authors provide video results for the driving sequences?**
>
> - We have already included a comprehensive set of video comparisons in the **supplementary material.**
> - To view them, please unzip the supplementary folder and open the `index.html` file in a web browser.
>     - These videos showcase the temporal consistency and high geometric fidelity of our reconstructions across various driving sequences.
>     - We believe these results strongly support our paper's claims.
>
> ---
>
> #### **Q3: Could you provide some qualitative examples of these failure cases (extremely complex, fast-moving, or highly non-rigid objects)?**
>
> Cause we couldn't upload PDFs. We claim the mechanism for ****Failure cases **as follows:**
>
> - Our method relies on LiDAR data to provide strong geometric supervision.
>     - In these extreme cases, the sparse LiDAR points may not align perfectly with the object's appearance in the camera image due to the motion blur.
> - When LiDAR points for fast-moving dynamic objects poor, the optimization process may prioritize minimizing the 2D photometric error.
>     - It may cause it to "bake" the motion blur artifact into the rendered texture instead of reconstructing a crisp 3D shape.
>     - This can be considered a form of overfitting to a transient, motion-induced visual artifact.
> - We will add a dedicated section with qualitative examples of these failure cases to the supplementary material in the next version.
>
> ---
>
> #### **Q4: Adding finer grids can harm geometric performance.**
>
> - Our ablation results in **Table 5** validate our hierarchical design by demonstrating that performance depends on a balanced coarse-to-fine structure, not merely on grid resolution.
> - The experiments show that:
>     - Both naively adding excessively fine grids (which overfits photometric noise) and removing the essential coarse grid (which destabilizes optimization) significantly harm geometric accuracy.
>     - Respectively. Our full three-level model achieves the best balance and performance with a CD of **1.161**.
>
> ---
>
> #### **Q4: The robustness of fixed grid architecture & Does the optimal grid configuration change per-scene (e.g., night vs. day), requiring manual tuning?**
>
> - Crucially, our architecture is robust and does not require per-scene tuning.
> - We used the **identical three-level architecture and hyperparameters** across all four diverse datasets—which included varied conditions like day, night, and different sensors—and achieved consistent results.
> - This proves the design generalizes well without manual adjustments.
>
> ---
>
> #### **Q5: The method performs appearance correction in 2D as a post-processing step.**
>
> - Our method offers a pragmatic and efficient alternative to more complex 3D disentanglement techniques.
> - It is not a simple post-processing step but is **deeply integrated into the reconstruction pipeline**.
> - Through a joint optimization objective, the 2D appearance transformations are guided by the rendered 3D scene and constrained by 3D LiDAR data, ensuring they remain geometrically consistent.
>
> ---
>
> #### **Q5: How does this compare conceptually to methods that disentangle appearance in 3D (e.g., Relightable 3DGS)?**
>
> - Our methods:
>     - The 2D approach we proposed is not only more efficient and easier to plug into other methods, but also boasts strong practicality and feasibility in real-world applications.
>     - Our approach can be easily integrated into various reconstruction and neural rendering pipelines to improve their reconstruction quality, without additional modifications to the representation.
>         - as demonstrated in our comparisons with OmniRe, ChatSim, and StreetGS.
> - The 3D approaches:
>     - The 3D approach is capable of creating a physical representation of the object/environment
>     - 3D approach e.g., Relightable 3DGS can better model the underlying physics of 3D view-dependent effects by scene properties like roughness, and albedo.
>         - But they usually need additional modifications to the representation.
>     - For large-scale, in-the-wild driving scenarios, accurately modeling these properties for all scene elements is a challenging, under-constrained problem.
>     - Often significantly more complex and computationally intensive.
> - We believe our method offers a valuable and practical solution for a critical problem in autonomous driving, while we appreciate the reviewer's insight and full 3D disentanglement remains a worthwhile but distinct future research direction.
>
> ---
>
> #### **Q5: What are the fundamental limitations of a 2D approach for handling 3D view-dependent effects like specular highlights?**
>
> - **Limitations**: It cannot model the underlying physics of 3D view-dependent effects.
> - For a specular highlight, it does not create a physically accurate 3D representation of the highlight that would behave correctly from all novel viewpoints.
> - However, our method has the ability to **effectively mitigate their negative impact** on geometric reconstruction.
>
> ---
>
> We appreciate all your constructive comments and we will improve our work according to them in the near future. We respectfully hope that this clarification will prompt you to reconsider your assessment and potentially raise the score of our submission. If the reviewer has any follow-up questions, we are happy to discuss them.
>
> > Chen, Ziyu, et al. "Omnire: Omni urban scene reconstruction." arXiv preprint arXiv:2408.16760 (2024).

---

> > ### Comment · Reviewer_2hvk · 2025-08-04
> >
> > Thank you for your rebuttal and the clarifications.
> >
> > I still have a couple of follow-up questions, mainly concerning the generalization and scope of your proposed strategy:
> >
> > (1) Your experiments convincingly show the method's effectiveness in per-scene optimization paradigms (e.g., OmniRe). However, the input mismatch is also a fundamental challenge for feed-forward frameworks like STORM[1], which aim to generalize across unseen scenes. Could you elaborate on how you envision your strategy performing in such a setting? My main concern is whether the solution is inherently tailored to per-scene adaptation, or if it can truly generalize.
> >
> > (2) Claiming to effectively “solve” this foundational problem of mismatch/inconsistency seems rather strong, especially considering that other paradigms such as feed-forward frameworks (e.g., STORM [1]) and generative models (e.g., MagicDrive [2]) can also perform reconstruction or novel view synthesis, and they also face issues related to input mismatch and inconsistency. Your validation appears to be limited to the per-scene reconstruction paradigm. I am wondering if positioning the work as a highly effective solution within the per-scene optimization paradigm would be a more precise characterization of its core contribution.
> >
> >
> > **References:**
> >
> > - Yang, Jiawei, et al. STORM: Spatio-temporal reconstruction model for large-scale outdoor scenes."
> >
> > - Gao, Ruiyuan, et al. MagicDrive: Street view generation with diverse 3D geometry control.

---

> > > ### Author Response · Authors · 2025-08-06
> > >
> > > We appreciate your comments, and address your concerns as follows:
> > >
> > > ---
> > >
> > > > (1) Envision strategy performing in feed-forward settings.  Concerns for generalization.
> > > >
> > > - We think the proposed multi-scale bilateral grid has a **clear path** toward integration into feed-forward frameworks. Our reasoning is as follows:
> > >     - **Existing feed-forward models already use simpler solutions.** As the reviewer points out, feed-forward frameworks also face appearance inconsistencies.
> > >         - STORM [A] incorporates appearance codes (3.2 "auxiliary tokens for sky and exposure") to mitigate this.
> > >         - Our work can be seen as a more powerful and expressive replacement, capable of handling complex, spatially-varying inconsistencies.
> > >     - **Bilateral grids can be feed-forward predicted.** There is strong precedent for predicting bilateral grid coefficients in a feed-forward manner from a single image observation.
> > >         - HDRNet [B] has used deep networks to predict the affine parameters of a bilateral grid for real-time image enhancement.
> > >         - Work in photorealistic style transfer [C] predicts an affine bilateral grid for style transfer.
> > > - Based on this, we envision that our multi-scale bilateral grid could be integrated into a feed-forward model by training a predictive module.
> > >     - This predictive module would learn to map image features (e.g., from a CNN backbone) to the hierarchical photometric transformations our grid represents.
> > >     - This would allow a model like STORM to correct for local variations, thus improving geometric consistency and rendering quality for unseen scenes.
> > >
> > > ---
> > >
> > > > (2) Claiming seems rather strong. Validation appears to be limited to the per-scene reconstruction paradigm.
> > > >
> > > - We thank the reviewer for your thoughtful feedback.
> > > - We have checked our manuscript and confirm that we use terms like "address" (e.g., Lines 61, 125, 156, 310) rather than "solve.". We can further revise some instances to "alleviate" or "mitigate."
> > > - We agree that characterizing our contribution as "**a highly effective solution for addressing photometric inconsistency within the per-scene optimization paradigm**" is a precise characterization. We will revise our claims to reflect this.
> > > - However, we would also like to respectfully highlight the our contribution for the broader field of 3D reconstruction beyond this specific paradigm:
> > >     - Our work provides **a critical component for explicit 3D representations.** And the need for an appearance model is fundamentally tied to 3D representation:
> > >         - **Generative models** (e.g. MagicDrive [D]), which operate in the 2D image space via diffusion, can implicitly handle inconsistencies within the diffusion model itself.
> > >         - **Feed-forward models** (e.g. STORM [A]), which rely on explicit 3D representations, require an explicit mechanism for appearance modeling. (affine tokens)
> > >     - **Generative models** can integrate with **per-scene optimization** methods in autonomous driving simulation.
> > >         - Recondreamer [E], Difix3D+ [F] represent the scene with diffusion and Gaussian Splatting hybird models.
> > >         - Appearance modelling is also important to reduce the artifacts caused by floaters.
> > >     - Per-scene optimization is still valuable, even for feed-forward models.
> > >         - Per-scene optimization do better at precisely recompute for specific scenes or assets.
> > >         - For some feed-forward methods, e.g. PartRM [G], per-scene optimization methods can provide GT reconstruction for supervision.
> > >
> > > Therefore, while we will adopt the reviewer's suggested framing, we believe our method is a critical component that enhances how any **explicit 3D representation** handles real-world photometric variations.
> > >
> > > ---
> > >
> > > Thank the reviewer again for your valuable comments. We have cited STORM and Magicdrive in our manuscripts in related work section 2.2 *"Autonomous driving simulation has emerged as a critical tool......by generating diverse, realistic driving scenarios, such as [Gao et al. 2023 MagicDrive], [Yang et al. 2024 STORM]"* .
> > >
> > > We will expand our discussion to better contextualize our work with respect to feed-forward and generative approaches, framing this as a promising direction for future work.
> > >
> > > ---
> > >
> > > > [A] Yang, Jiawei, et al. Storm: Spatio-temporal reconstruction model for large-scale outdoor scenes.
> > > >
> > > >[B] Gharbi, Michaël, et al. Deep bilateral learning for real-time image enhancement.
> > > >
> > > >[C] Xia, Xide, et al. Joint bilateral learning for real-time universal photorealistic style transfer.
> > > >
> > > >[D] Gao, Ruiyuan, et al. MagicDrive: Street view generation with diverse 3D geometry control.
> > > >
> > > >[E] Ni, Chaojun, et al. Recondreamer: Crafting world models for driving scene reconstruction via online restoration.
> > > >
> > > >[F] Wu, Jay Zhangjie, et al. Difix3d+: Improving 3d reconstructions with single-step diffusion models.
> > > >
> > > >[G] Gao, Mingju, et al. PartRM: Modeling Part-Level Dynamics with Large Cross-State Reconstruction Model.
> > > >

---

> ### Author Response · Authors · 2025-08-08
> **Regarding Paper 1353: Author-Reviewer Discussion Approaching The End**
>
> Dear Reviewer 2hvk,
>
> We greatly appreciate your time and effort in reviewing our submission.
>
> As the discussion period approaches its end, we just wanted to kindly check if our rebuttal has sufficiently addressed your concerns. Your comments are very valuable to us, and we look forward to hearing from you.
>
> Thank you again for your time.
>
> Best regards,
>
> The Authors of Paper 1353

---

> > ### Comment · Reviewer_2hvk · 2025-08-09
> >
> > Thank you for your rebuttal and further clarification. I will carefully consider all reviewers’ comments when updating my final score.

---

### Official Review · Reviewer_yZ4i · 2025-07-02

**Clarity:** 3
**Significance:** 3
**Originality:** 2
**Rating:** 4
**Confidence:** 4

**Summary:**

This paper proposes multi-scale bilateral grids to address the photometric inconsistency in a temporal multi-camera setting. The proposed method combines the advantages of appearance code and vanilla bilateral grid, thus being optimization-friendly and having better modeling capacity. The paper conducts extensive experiments on four driving datasets, demonstrating good performance.

**Questions:**

1. Why are the performance improvements on the Argoverse and Pandaset datasets less prominent compared with those on the Waymo and nuScenes datasets, including both the 3D reconstruction and novel view synthesis tasks?
2. How is the performance of the baseline model with a smaller sized single-scale bilateral grid, e.g., with only the finest bilateral grid in the proposed multi-scale setting?

**Ethical Concerns:**

["NO or VERY MINOR ethics concerns only"]

**Final Justification:**

Thanks for the thoughtful response.
W1: The explanation about novelty focuses on the unification of two paradigms and indeed addresses my concern.
W2 & Q1: The results of additional experiments are impressive and demonstrate the advantages of the proposed method in extreme cases.
Q2: The comparison with the single-scale baseline validates the effectiveness of the multi-scale design.
Generally, the rebuttal has addressed all of my concerns and I will raise my score to 4. It is recommended to integrate rebuttal experiments and analysis into the revised version.

**Limitations:**

Yes

**Paper Formatting Concerns:**

None.

**Quality:**

3

**Strengths And Weaknesses:**

Strengths:
1. Good motivation. The photometric inconsistency is an important problem in 3D reconstruction. The paper proposes to address it by combining the advantages of appearance code and bilateral grids.
2. Good writing. The paper is well written and easy to follow.

Weaknesses:
1. Lack of novelty. The extension of single-scale bilateral grid to the multi-scale counterpart is quite straightforward and does not reflect much insight.
2. Inconsistency in performance improvement. The performance improvements on the Argoverse and Pandaset datasets are less prominent and even minimal compared with those on the Waymo and nuScenes datasets. The paper does not include any discussion on it.

---

> ### Author Rebuttal · Authors · 2025-07-31
>
> We thank **yZ4i** for the feedback and for recognizing our good motivation and clear writing. We address the concerns about novelty and performance consistency below.
>
> ---
>
> #### **W1: Lack of novelty & does not reflect much insight.**
>
> - We respectfully disagree with the assessment that our work lacks insight.
>     - Our primary contribution is not the simple extension to a multi-scale structure, but rather the insightful **unification of two distinct paradigms**: global appearance codes and pixel-wise bilateral grids.
> - Our framework has a principled design that provides significant advantages by addressing key limitations in prior work:
>     - **Optimization Challenges**: Our method overcomes the critical optimization challenges and instability inherent to single high-resolution bilateral grids by decomposing the complex transformation across multiple, more manageable scales.
>     - **Hierarchical Refinement**: We introduces a **hierarchical, residual refinement** process where the coarse grid learns a global base appearance (akin to an appearance code), while finer grids learn residual corrections for regional and local details. This structured, coarse-to-fine learning process is a key insight that enables stable optimization.
>     - **Bridges Global and Local Modeling**: Our approach provides the localized, pixel-wise adjustment that global appearance codes lack, while maintaining the global consistency and optimization stability that single bilateral grids struggle with.
> - Therefore, the novelty also lies not in how we leverage this structure to create a unified and robust framework that solves core limitations of existing methods.
>
> ---
>
> #### **W2 & Q1: Improvements on the Argoverse and Pandaset datasets less prominent**
>
> - The performance variation across datasets stems from **a. the characteristics of the test scenes** and **b. the intrinsic capabilities of Gaussian Splatting**.
>     - We would like to clarify that our method provides a consistent and crucial improvement in **geometric accuracy** across all tested conditions.
>     - **Dataset Characteristics.** In our evaluation, the selected test sequences from the Argoverse and PandaSet datasets feature relatively more consistent lighting conditions.
>         1. Since our method's primary function is to resolve appearance inconsistencies, the margin for improvement on appearance-based metrics is naturally smaller in these scenarios.
>     - **Gaussian Splatting has capabilities of fitting appearance variance using "floater".**
>         1. It can model some inconsistencies by adjusting its Spherical Harmonics (SH) coefficients and by using semi-transparent Gaussians.
>         2. This allows the baseline to achieve a reasonable appearance match in less challenging scenes, which is why the PSNR/SSIM improvement of our method is more modest on those datasets.
>         3. However, this baseline behavior of **"absorbing" appearance errors into the Gaussian representation** often comes at the **expense of geometric accuracy**.
> - Our method resolves these appearance inconsistencies first, which enables the optimizer to achieve a more accurate geometric reconstruction, reducing the Chamfer Distance on Argoverse by **15%** (from 0.954 to 0.807) and on PandaSet by **10%** (from 0.503 to 0.453) over the baseline.
> - **Additional experiments to validate.** We conduct additional experiments on a curated subset of scenes featuring more extreme lighting and weather conditions. The scenes are listed here:
>
> > Table R3: Additional challenging Argoverse scenarios.
> |scene ID|description |
> | --- | --- |
> | 49e970c4 | Extreme Lighting, Overexposure, Lens Flare |
> | 8184872e | Complex Shadows, Direct Sunlight, Information Loss |
> | 91923e20 | Extreme Lighting, Camera Artifacts. |
> | a8a2fbc2 | Low-Angle Sun, Texture Disentanglement. |
> | b403f8a3 | Overpass Structure, Extreme Lighting Transitions|
> | eaaf5ad3 | Multiple Artificial Lights, Wet Surfaces, Rainy Conditions |
>
> > Table R4: Evaluation results of our multi-scale bilateral grids on additional challenging Argoverse scenarios:
> |  | Recon. PSNR$\uparrow$ | Recon. SSIM$\uparrow$ | Recon. CD$\downarrow$ | Recon. RMSE$\downarrow$ | Recon. Depth$\downarrow$ | NVS PSNR$\uparrow$ | NVS SSIM$\uparrow$ |
> | --- | --- | --- | --- | --- | --- | --- | --- |
> | 49e970c4 | 23.35 | 0.789 | 1.229 | 5.175 | 0.060 | 21.33 | 0.673 |
> | 8184872e | 24.23 | 0.864 | 0.666 | 3.303 | 0.022 | 21.76 | 0.766 |
> | 91923e20 | 25.86 | 0.876 | 0.820 | 3.342 | 0.019 | 23.48 | 0.789 |
> | a8a2fbc2 | 25.13 | 0.823 | 0.774 | 3.513 | 0.033 | 22.65 | 0.716 |
> | b403f8a3 | 27.55 | 0.926 | 0.298 | 2.822 | 0.004 | 25.25 | 0.870 |
> | eaaf5ad3 | 25.60 | 0.899 | 0.566 | 3.228 | 0.022 | 23.28 | 0.824 |
> | Average | **25.29** | **0.863** | **0.726** | **3.564** | **0.027** | **22.96** | **0.773** |
>
> > Table R5: Evaluation results of OmniRe on additional challenging Argoverse scenarios:
> |  | Recon. PSNR$\uparrow$ | Recon. SSIM$\uparrow$ | Recon. CD$\downarrow$ | Recon. RMSE$\downarrow$ | Recon. Depth$\downarrow$ | NVS PSNR$\uparrow$ | NVS SSIM$\uparrow$ |
> | --- | --- | --- | --- | --- | --- | --- | --- |
> | 49e970c4 | 21.99 | 0.772 | 1.474 | 5.270 | 0.129 | 20.37 | 0.659 |
> | 8184872e | 22.79 | 0.850 | 0.862 | 3.389 | 0.101 | 20.79 | 0.751 |
> | 91923e20 | 24.47 | 0.859 | 1.171 | 3.463 | 0.124 | 22.67 | 0.776 |
> | a8a2fbc2 | 23.13 | 0.805 | 1.107 | 3.602 | 0.140 | 21.20 | 0.701 |
> | b403f8a3 | 26.65 | 0.922 | 0.396 | 2.862 | 0.014 | 24.77 | 0.869 |
> | eaaf5ad3 | 24.69 | 0.889 | 0.776 | 3.295 | 0.083 | 22.79 | 0.810 |
> | Average | 23.95 | 0.850 | 0.964 | 3.647 | 0.098 | 22.10 | 0.761 |
>
> In these challenging Argoverse scenarios, our method outperforms OmniRe by an average of **1.338 dB** in PSNR.
>
> > Table R6: Additional challenging Pandaset scenarios.
> |scene ID|description |
> | --- | --- |
> | 19 | Low-Angle Sun, Hard Shadows |
> | 21 | Low-Angle Sun, High Dynamic Range |
> | 29 | Low-Angle Sun, Deep Shadows |
> | 48 | Low-Angle Sun, High Dynamic Range |
> | 52 | Low-Angle Sun, Sun Glare |
> | 63 | Night Scene, Multiple Light Sources, Glare |
>
> > Table R7: Evaluation results of our multi-scale bilateral grids on additional challenging Pandaset scenarios:
> |  | Recon.  PSNR$\uparrow$ | Recon.  SSIM$\uparrow$ | Recon.  CD$\downarrow$ | Recon.  RMSE$\downarrow$ | Recon.  Depth$\downarrow$ | NVS PSNR$\uparrow$ | NVS SSIM$\uparrow$ |
> | --- | --- | --- | --- | --- | --- | --- | --- |
> | 19 | 24.97 | 0.812 | 0.639 | 5.697 | 0.040 | 21.94 | 0.645 |
> | 29 | 27.95 | 0.864 | 0.274 | 2.559 | 0.012 | 23.94 | 0.725 |
> | 48 | 28.30 | 0.878 | 0.260 | 3.026 | 0.020 | 24.21 | 0.709 |
> | 21 | 26.98 | 0.849 | 0.947 | 4.886 | 0.024 | 22.91 | 0.686 |
> | 52 | 28.87 | 0.870 | 0.557 | 3.737 | 0.023 | 24.83 | 0.727 |
> | 63 | 30.20 | 0.902 | 0.755 | 3.801 | 0.014 | 27.49 | 0.853 |
> | Average | **27.88** | **0.863** | **0.572** | **3.951** | **0.022** | **24.22** | **0.724** |
>
> > Table R8: Evaluation results of OmniRe on additional challenging Pandaset scenarios:
> |  | Recon.  PSNR$\uparrow$ | Recon.  SSIM$\uparrow$ | Recon.  CD$\downarrow$ | Recon.  RMSE$\downarrow$ | Recon.  Depth$\downarrow$ | NVS PSNR$\uparrow$ | NVS SSIM$\uparrow$ |
> | --- | --- | --- | --- | --- | --- | --- | --- |
> | 19 | 24.60 | 0.811 | 0.771 | 5.702 | 0.045 | 21.78 | 0.643 |
> | 29 | 27.61 | 0.863 | 0.313 | 2.566 | 0.016 | 23.85 | 0.726 |
> | 48 | 27.70 | 0.873 | 0.288 | 3.031 | 0.027 | 23.96 | 0.705 |
> | 21 | 26.35 | 0.842 | 1.035 | 4.900 | 0.035 | 22.69 | 0.679 |
> | 52 | 28.06 | 0.868 | 0.583 | 3.757 | 0.030 | 24.44 | 0.726 |
> | 63 | 29.35 | 0.898 | 0.812 | 3.820 | 0.021 | 26.74 | 0.845 |
> | Average | 27.28 | 0.859 | 0.634 | 3.963 | 0.029 | 23.91 | 0.721 |
>
> In these challenging Pandaset scenarios, our method outperforms OmniRe by an average of **0.6 dB** in PSNR, showing significant improvements when facing scenarios with obvious appearance inconsistencies.
>
> ---
>
> #### **Q2: How is the performance of the baseline model with a smaller sized single-scale bilateral grid?**
>
> - **Smaller sized bilateral grids do not lead to better results.** Motivated by reviewer **yZ4i's** suggestion, we ran a **new experiment** using only the finest grid from our hierarchy $(8\times 8\times 4\times 12)$ as a single-scale baseline.
>
> > Table R9: Additional ablation study on grid size.
> |  | Recon. PSNR$\uparrow$ | NVS PSNR$\uparrow$ | CD$\downarrow$ |
> | --- | --- | --- | --- |
> | Ours | **27.69** | **24.64** | **1.161** |
> | OmniRe w/single grid (16,16,8) | 25.98 | 23.60 | 1.380 |
> | OmniRe w/single grid (8,8,4) | 26.56 | 23.90 | 1.376 |
> | OmniRe w/grid (8,8,4)+(2,2,1) | 27.57 | 24.57 | 1.340 |
> - The single fine-scale grid performs poorly compared to our full model.
>
> The central premise of our work is that single-scale bilateral grids, especially fine ones, are difficult to optimize for complex scenes because they lack a stable, global initialization.
> - Our ablations in **Table 5(b)** ("Grid Size Combinations") also provide evidence for this.
>     - Removing the essential coarse grid (the `(4,4,2)+(8,8,4)` configuration) causes a severe degradation in geometric performance, with the Chamfer Distance (CD) increasing from 1.161 to **1.963**.
>     - This shows that without the coarse grid providing a stable base transformation, the finer grids fail to converge properly, a challenge we also illustrate qualitatively in **Figure 1(b)** and **Figure 4(b)**.
> ---
>
> We appreciate all your constructive comments and we will improve our work according to them in the near future. We respectfully hope that this clarification will prompt you to reconsider your assessment and potentially raise the score of our submission. If the reviewer has any follow-up questions, we are happy to discuss them.

---

> > ### Comment · Reviewer_yZ4i · 2025-08-06
> >
> > Thanks for the thoughtful response.
> > W1: The explanation about novelty focuses on the unification of two paradigms and indeed addresses my concern.
> > W2 & Q1: The results of additional experiments are impressive and demonstrate the advantages of the proposed method in extreme cases.
> > Q2: The comparison with the single-scale baseline validates the effectiveness of the multi-scale design.
> > Generally, the rebuttal has addressed all of my concerns and I will raise my score to 4. It is recommended to integrate rebuttal experiments and analysis into the revised version.

---

### Official Review · Reviewer_5GsZ · 2025-07-03

**Clarity:** 3
**Significance:** 3
**Originality:** 3
**Rating:** 5
**Confidence:** 3

**Summary:**

This paper proposes a method to learn to represent the appearance variation in autonomous driving scenes reconstruction with 3D gaussian splatting scene graph.
Instead of relying on a single appearance embedding that provides a global transform on the image level but fails to represent local variation, or a bilateral grid that will learn pixel-wise variation but is hard to optimize, this paper proposes to use a composition of multi-scale bilateral grids.
The coarse bilateral grid will ressemble the appearance code by  modeling an image level transformation, then progressively  reduce patch size to learn more local transformations.
By doing so, the model is able to learn more local variation than appearance embedding, while also guiding the bilateral grid training with a composition of progressively more detailed transformation.
By doing so, this paper is able to provide better geometric and photometric results than using either appearance code or bilateral grid, as it is shown through extensive benchmarking on 4 different driving datasets.

**Questions:**

How would this method perform in very difficult scenarios (night scenes, sun flare)?
Are there cases where the multi-scale bilateral grids are able to overfit to some dynamic element instead of representing it correctly in 3D?

**Ethical Concerns:**

["NO or VERY MINOR ethics concerns only"]

**Final Justification:**

My questions were answered. I think it is overall a good paper with interesting contributions.

**Limitations:**

Yes

**Quality:**

3

**Strengths And Weaknesses:**

Strengths:
- SOTA reconstruction quality compared to appearance code and bilateral grids.
- The idea is simple and intuitive yet provides good results.
- Paper is overall clear, with benchmarking on 4 different driving datasets.
- Significantly less parameters to process compared to purely pixel-wise bilateral grids.

Weaknesses:
- Computation is much higher compared to simply using appearance code.

---

> ### Author Rebuttal · Authors · 2025-07-31
>
> We thank **5GsZ** for the positive evaluation and for recognizing our state-of-the-art reconstruction quality and the efficiency of our approach compared to pixel-wise bilateral grids.
>
> ---
>
> #### **W1: Computation is much higher compared to simply using appearance code.**
>
> **We focus on favorable trade-off.** We designed the framework to ensure this overhead is moderate and justified by the significant improvements in reconstruction quality.
>
> - **Modest Computational Cost.** Our approach increases training time by only **9%** over the AC baseline (from 1.93 to 2.10 hours).
> - **Substantial Performance Gain.** In return for this modest overhead, our method achieves a **19% reduction in geometric error** on the NuScenes dataset (**Table 1**), with the Chamfer Distance dropping from 1.437 to **1.161**.
> - We believe this demonstrates a valuable trade-off, where a small increase in computation yields a substantial and critical improvement in geometric accuracy, which is crucial for autonomous driving applications.
>
> ---
>
> #### **Q1: How would this method perform in very difficult scenarios (night scenes, sun flare)?**
>
> Our results show that the framework can work well and demonstrate even **greater advantages in these challenging environments.**
>
> - **Qualitative Results. Figure 3(e)** and the **left panel of** **Figure 4** provide qualitative visualizations of our method's ability to produce high-fidelity, coherent reconstructions in low-light environments. Sun flare is also included in our test scenarios. We will add more examples in our final version.
> - **Quantitative Improvement.** We select a **subset** consisting of scenarios featuring **night scenes and sun flare** to make further evaluation. Our method shows a marked quantitative improvement over baseline approaches.
>
> > Table R1: Additional comparison results on Night Scenes: waymo-10, waymo-30, argoverse-eaaf5ad3, pandaset-63.
> |  | Recon.  PSNR$\uparrow$ | Recon.  SSIM$\uparrow$ | Recon.  CD$\downarrow$ | Recon.  RMSE$\downarrow$ | Recon.  Depth$\downarrow$ | NVS PSNR$\uparrow$ | NVS SSIM$\uparrow$ |
> | --- | --- | --- | --- | --- | --- | --- | --- |
> | OmniRe | 28.34 | 0.805 | 0.586 | 2.767 | 0.041 | 26.54 | 0.753 |
> | Ours | **29.13** | **0.812** | **0.500** | **2.731** | **0.016** | **27.21** | **0.762** |
>
> > Table R2: Additional comparison results on Sun flare scenes: waymo-550, waymo-561, argoverse-49e970c4, argoverse-91923e20, argoverse-a8a2fbc2.
> |  | Recon. PSNR$\uparrow$ | Recon.  SSIM$\uparrow$ | Recon.  CD$\downarrow$ | Recon.  RMSE$\downarrow$ | Recon.  Depth$\downarrow$ | NVS PSNR$\uparrow$ | NVS SSIM$\uparrow$ |
> | --- | --- | --- | --- | --- | --- | --- | --- |
> | OmniRe | 26.68 | 0.866 | 0.781 | 3.073 | 0.079 | 24.98 | 0.796 |
> | Ours | **28.65** | **0.879** | **0.593** | **2.994** | **0.022** | **25.84** | **0.803** |
>
> ---
>
> #### **Q2: Are there cases where the multi-scale bilateral grids are able to overfit to some dynamic element instead of representing it correctly in 3D?**
>
> The reviewer's concerns are insightful. Actually it may happen without enough constrains, as we mentioned in our limitations.
>
> - Our framework prevent this type of overfitting by explicitly linking appearance modeling with **geometric constraints**.
>
> - As formulated in **Equation 4** . Our framework does not solely minimize photometric error; it simultaneously minimizes a **geometric loss** term against ground-truth LiDAR depth maps:
>
> > $\left\| D_{c,t}^r - D_{c,t} \right\|_2^2$.
> >
> - This design creates a powerful synergy between appearance and geometry:
>     - Any appearance transformation learned by the bilateral grids must also support a **geometrically accurate 3D reconstruction**.
>     - If the grid were to "overfit" to a transient visual effect (e.g., a shadow or reflection) by altering its appearance, it would likely create a **discrepancy with the ground-truth LiDAR data**.
>     - This discrepancy would incur a higher geometry loss, penalizing the model and guiding it to learn**corrections** **for true photometric inconsistencies** (like lighting changes) rather than incorrectly baking dynamic effects into the 3D geometry.
> - This process is a key factor in our method's ability to reduce geometric artifacts like "floaters" and achieve superior geometric fidelity across our experiments.
>
> ---
>
> We appreciate all your constructive comments and we will improve our work according to them in the near future. If the reviewer has any follow-up questions, we are happy to discuss them.

---

> > ### Comment · Reviewer_5GsZ · 2025-08-06
> >
> > Thank you, my questions have been answered

---

### Note · Authors · 2025-08-15

We thank all the reviewers for their constructive feedback on our work and are excited that they recognize the strengths of our work:

- State-of-the-art reconstruction quality and intuitive design (5GsZ).
- Good motivation in addressing a critical problem and clear writing (yZ4i).
- Novel and effective proposed method with thorough experiments (dvs1).
- Ability to tackle a critical challenge with strong quantitative results (2hvk).

We are also pleased that our rebuttal was well-received and we have addressed all the concerns raised by reviewers 5GsZ, yZ4i and dvs1. We successfully clarified:

- **Novelty** (yZ4i, 2hvk): We elaborated that our core contribution is the principled unification of global and local appearance modeling through a coarse-to-fine residual refinement, which solves critical optimization challenges inherent in prior methods.
- **Performance on Challenging Scenes** (yZ4i, dvs1, 5GsZ): We demonstrated that our method's advantages are even more pronounced in challenging scenarios.
- **Architecture Robustness** (2hvk): We confirmed that our fixed, three-level architecture is robust, requiring no per-scene tuning while delivering consistent state-of-the-art results across all four diverse datasets.
- **Analysis Rigor** (2hvk): We improved our initial histogram analysis with a more rigorous KL Divergence comparison against a ground-truth photometric inconsistency distribution, quantitatively proving our method better models real-world effects.

To conclude, we believe our work makes a significant contribution by presenting:

- A novel framework that unifies global and local appearance correction, overcoming the respective limitations of appearance codes and single-scale bilateral grids.
- A pragmatic and robust solution that delivers substantial improvements in geometric accuracy—a critical factor for autonomous driving safety—with only a modest computational overhead.
- A method proven to be highly effective and generalizable across four large-scale, diverse driving datasets without any manual tuning.

We will incorporate the valuable feedback, additional experiments, and refined analyses from our rebuttal into the final version of the paper.

Thank you once again for your time and thoughtful consideration. The discussion period has been invaluable in helping us clarify our contributions and strengthen our paper.

Best regards,

The Authors of Submission 1353

---

### Decision · Program_Chairs · 2025-09-17

**Decision:**

Accept (poster)

**Comment:**

The paper tackles inconsistent appearance in 3DGS driving scenes through a multi-scale bilateral grid that operates hierarchically to provide both global (coarse-level) and local (fine-level) corrections. Initial feedback of the reviewers was generally positive, highlighting the paper’s strengths while raising questions regarding performance in challenging scenarios, the degree of novelty, limited improvements on the Argoverse and Pandaset datasets, and the impact of using a smaller single-scale bilateral grid. The rebuttal successfully addressed these concerns, and overall the paper is considered strong, offering interesting and valuable contributions.